# How Well Can Differential Privacy Be Audited in One Run?

**Amit Keinan**    **Moshe Shenfeld**    **Katrina Ligett**
Department of computer science and engineering
The Hebrew University of Jerusalem
{amit.keinan2, moshe.shenfeld, katrina.ligett}@mail.huji.ac.il

## Abstract

Recent methods for auditing the privacy of machine learning algorithms have improved computational efficiency by simultaneously intervening on multiple training examples in a single training run. Steinke et al. [1] prove that one-run auditing indeed lower bounds the true privacy parameter of the audited algorithm, and give impressive empirical results. Their work leaves open the question of how precisely one-run auditing can uncover the true privacy parameter of an algorithm, and how that precision depends on the audited algorithm. In this work, we characterize the maximum achievable efficacy of one-run auditing and show that the key barrier to its efficacy is interference between the observable effects of different data elements. We present new conceptual approaches to minimize this barrier, towards improving the performance of one-run auditing of real machine learning algorithms.

## 1   Introduction

Differential privacy (DP) is increasingly deployed to protect the privacy of training data, including in large-scale industry machine learning settings. As DP provides a theoretical guarantee about the worst-case behavior of a machine learning algorithm, any DP algorithm should be accompanied by a proof of an upper bound on its privacy parameters. However, such upper bounds can be quite loose. Worse, analyses and deployments of differential privacy can contain bugs that render those privacy upper bounds incorrect. As a result, there is growing interest in *privacy auditing* methods that can provide empirical lower bounds on an algorithm's privacy parameters. Such lower bounds can help detect whether the upper bounds in proofs are unnecessarily loose, or whether there are analysis or implementation errors that render those bounds incorrect.

Differential privacy constrains how much a change in one training point is allowed to affect the resulting distribution over outputs (e.g., trained models). Hence, one natural approach to auditing DP, which we term "classic auditing," simply picks a pair of training datasets that differ in one entry and runs the learning algorithm over each of them repeatedly in order to discover differences in the induced output distributions. Estimating these distributions reasonably well (and hence obtaining meaningful lower bounds on the privacy parameters) requires hundreds or thousands of runs of the learning algorithm, which may not be practical. In response, there has been increasing interest in more computationally feasible auditing approaches that change multiple entries of the training data simultaneously. In particular, Steinke et al. [1] study privacy auditing with one training run (one-run auditing) and show impressive empirical results on DP-SGD.

Classic auditing is not only *valid* (informally: with high probability, the lower bounds on the privacy parameters it returns are indeed no higher than the true privacy parameters); it is also *asymptotically tight* (informally: there exists a pair of training datasets such that, if auditing is run for enough rounds, the resulting lower bounds approach the true privacy parameters). Steinke et al. [1] show that

one-run auditing (ORA) is also valid, but their work leaves open the question of how close ORA's lower bounds are to the true privacy parameters, and what aspects of the audited algorithm determine how tightly it can be audited in one run. We explore these questions in this work. Steinke et al. [1] empirically demonstrate that one-run auditing of specific algorithms seems to not be asymptotically tight. Indeed, our work confirms this suspicion.

We study guessing-based auditing frameworks, where a lack of privacy is demonstrated by a guesser's ability to correctly guess, based on an algorithm's output, which input training points (from a set of known options) generated the output. As such, we are interested in the *efficacy* of auditors (informally, their expected ratio of correct guesses to guesses overall) as a measure of their ability to uncover an algorithm's true privacy parameters. Auditors are allowed to *abstain* from guessing about some points. We study auditors both with and without abstentions to pinpoint the role of abstentions.

The performance of all auditing methods is influenced by the choice of the auditing datasets and the guessing method, and sub-optimal choices lead to loose lower bounds. We focus on the inherent limitations of ORA, even under optimal choice of these parameters.

We focus on auditing pure $\varepsilon$-DP. Steinke et al. [1] also study auditing approximate $(\varepsilon, \delta)$-DP (bounding $\varepsilon$, given some $\delta$). We choose this simpler setting to zoom in on the fundamental limitations of ORA that appear even when $\delta = 0$; we expect our findings to remain relevant for approximate DP. In the common regime where $\delta$ is very small, the observed behavior of an approximate DP algorithm is very similar to a pure DP counterpart with a similar $\varepsilon$,[1] and in particular the efficacy of an auditor for the approximate DP algorithm is very similar to that for its pure counterpart. Moreover, observed high privacy loss of an approximate DP algorithm could result from the $\delta$-tail of the privacy loss distribution, weakening the bounds on $\varepsilon$ that an approximate DP auditor can infer. Thus, auditing approximate DP may only create an additional gap (compared to auditing pure DP) for all auditing methods. Steinke et al. [1, Section 7] discuss this gap for ORA and explore it empirically (Section 7).

## 1.1 Our Contributions

In Section 4, we show that ORA's efficacy is fundamentally limited—there are three fundamental gaps between what it can discover and the true privacy parameters, which we illustrate by three simple algorithms. ORA fails to detect the true privacy parameters if: (1) the algorithm provides poor privacy to a small subset of the input elements, (2) the algorithm only rarely produces outputs that significantly degrade privacy, or (3) the algorithm's output inextricably mixes multiple input elements, making it difficult to isolate individual effects when multiple elements are being audited simultaneously. In Theorem 5.2 we give a characterization of the optimal efficacy of ORA, formalizing the three gaps and showing that they are exactly the gaps of ORA. In Theorem 5.3 we use this result to characterize the algorithms for which ORA is asymptotically tight. These are the algorithms that sufficiently often realize their worst-case privacy loss in a way that can be isolated per training point. In addition, we show parallel characterizations of optimal efficacy and asymptotic tightness for guessers that are required to guess for every element, clarifying the role of abstention (see Theorems 5.1 and C.10).

In Section 6 we explore, both theoretically and empirically,[2] auditing of the most important DP algorithm for learning, DP-SGD, as a case study of ORA. While versions of gaps (1) and (2) exist for DP-SGD, gap (3)—which we refer to as the interference gap—in particular still looms large. We explore the common approach to mitigate this gap and two new conceptual approaches we propose, including a new adaptive variant of ORA, to further mitigate it and improve one-run auditing.

## 1.2 Related Work

Privacy auditing is often applied to privacy-preserving machine learning algorithms using membership inference attacks [3], where differences in the induced distributions over outputs under differing training data enable an auditor to guess some of the training points [3–5]. Jagielski et al. [6] suggest a membership inference attack that is based on the loss of the model on the element. Nasr et al. [7]

---

[1] Kasiviswanathan and Smith [2] state and prove such a claim formally (Lemma 3.3, Part 2) (note that the journal version had a typo in the claim, which was corrected in the arXiv version). The lemma is stated for a pure counterpart with privacy level of $2\varepsilon$, but can be extended to arbitrary privacy level $> \varepsilon$ with a different blow-up term in $\delta$.

[2] Code for running the experiments is available at `https://github.com/amitkeinan1/exploring-one-run-auditing-of-dp`.

suggest exploiting the gradients from the training process to conduct stronger attacks. Jagielski et al. [6] introduce such methods to lower-bound the privacy level of an algorithm, and demonstrate it for DP-SGD [8]. This method achieves asymptotically tight bounds when equipped with optimal datasets and attacks, but is computationally burdensome.

Malek Esmaeili et al. [9] suggest a significantly more efficient auditing method that conducts a membership inference attack on multiple examples simultaneously in a single run of the algorithm, later evaluated more rigorously by Zanella-Beguelin et al. [10]. Steinke et al. [1] prove that this method is valid. They show that this method produces asymptotically tight bounds for local randomized response, and suggest it may be inherently limited for other algorithms.

Some works use the notion of f-DP (a generalization of differential privacy) for tighter auditing [7, 11]. In a contemporaneous work, Xiang et al. [12] also develop an f-DP-based method for auditing in one run which uses an information-theoretic perspective. Their work also takes note of the interference gap that we study (we have adapted their term "interference"), but does not provide the complete characterization of the gaps of ORA or ways to handle this gap as we do.

## 2  Preliminaries

We study the auditing of algorithms that operate on ordered datasets consisting of $n$ elements from some universe $X$.[3] Given a randomized algorithm $M : X^n \to \mathcal{O}$, any dataset $D$ induces a distribution over outputs $M(D)$ of the algorithm. Differential privacy [13] bounds the max-divergence (see Definition C.2) between output distributions induced by neighboring datasets; datasets $D, D' \in X^n$ are neighboring if $|\{i \in [n] : D_i \neq D_i'\}| \leq 1$. In this case, we write $D \simeq D'$.

**Definition 2.1** (Differential Privacy (DP) [13]). The differential privacy level of a randomized algorithm $M : X^n \to \mathcal{O}$ is $\varepsilon(M) := \sup_{D \simeq D' \in X^n} D_\infty(M(D)||M(D'))$. $M$ is $\varepsilon$-differentially private if its privacy level is bounded by $\varepsilon$, that is, if $\varepsilon(M) \leq \varepsilon$.

For simplicity, we focus our analysis on algorithms for which the supremum is a maximum,[4] i.e., there exist $D \simeq D' \in X^n$ such that $D_\infty(M(D)||M(D')) = \varepsilon(M)$.

*One-run auditing* [1] bounds the privacy level of a given algorithm according to the success in a guessing game. The one-run auditor (see Algorithm 1) gets oracle access to an algorithm $M : X^n \to \mathcal{O}$ to audit, and takes as input a pair vector and a guesser that define its strategy. The pair vector $Z = (x_1, y_1, ..., x_n, y_n) \in X^{2n}$ is a pair of options for each entry of the dataset on which we will audit $M$,[5] and the guesser $G : \mathcal{O} \to \{-1, 0, 1\}^n$ is a function that defines how the auditor makes guesses based on the algorithm's output.

The one-run auditor runs a game in which it samples a random vector $S \in \{-1, 1\}^n$ uniformly, and uses it to choose one element from each pair in $Z$ to define the dataset $D$. Then, it feeds $D$ to $M$ to get an output $o$. Based on the output $o$, the auditor uses the guesser $G$ to output a vector of guesses $T$ for the random bits of $S$, where $T_i$ is the guess for the value of $S_i$ and a value $T_i = 0$ is interpreted as an abstention from guessing the $i$th element. The auditor outputs a pair of numbers $(v, r)$: the number of correct guesses (that is, the number of indexes in which the guesses in $T$ are equal to the random bits in $S$: $v := |\{i \in [n] : T_i = S_i\}|$) and the number of taken guesses (that is, the number of non-zero indexes $r := |\{i \in [n] : T_i \neq 0\}|$).[6] When the algorithm to audit and the strategy are clear from context, we denote the random variables for $(v, r)$ as $(V_n, R_n)$.

Steinke et al. [1] prove that these two counts yield a lower bound with confidence level $1 - \beta$ on the true privacy level $\varepsilon(M)$ of $\varepsilon_\beta'(v, r) := sup(\{\varepsilon \in \mathbb{R}^+ : Pr[\text{Bin}(r, p(\varepsilon)) \geq v] \leq \beta\})$, where $p$ is the standard logistic function $p(x) := \frac{e^x}{e^x + 1}$ (extended with $p(-\infty) := 0$ and $p(\infty) := 1$).

---

[3]Notice that algorithms over ordered datasets are more general than algorithms over unordered datasets.

[4]Notice that even if the supremum is not achieved for $M$, for every $a > 0$, there exist $D \simeq D' \in X^n$ such that $D_\infty(M(D)||M(D')) \geq \varepsilon(M) - a$, and hence this assumption does not weaken our results.

[5]Steinke et al. [1] focus on the ORA variant where $Z$ is a vector of elements to include or exclude, while we consider the variant where $Z$ is a vector of pairs of options (this allows the analysis of algorithms over ordered datasets which are more general than algorithms over unordered datasets; our results naturally extend to the other variant). In Steinke et al. [1], the adversary can choose to fix some rows (to allow simultaneously auditing and training on real data); our analysis covers this option by considering the fixed rows as part of the algorithm.

[6]We assume that $r > 0$ because an auditor cannot benefit from not making any guesses.

# 3 Problem Setting

In this section, we lay the formal foundation for analyzing the efficacy and tightness of ORA. Appendix A extends this to general guessing-based audit methods.

We say that ORA is asymptotically tight for an algorithm if there exists a strategy such that the number of taken guesses approaches $\infty$ and the bounds approach the privacy level of the algorithm.

**Definition 3.1** (ORA Asymptotic Tightness). ORA is asymptotically tight for a randomized algorithm $M : X^* \to \mathcal{O}$ if there exists a strategy $\{(Z_n, G_n)\}_{n \in \mathbb{N}}$ with unlimited guesses, i.e., $R_n \xrightarrow[n \to \infty]{P} \infty$,[7] such that for every confidence level $0 < 1 - \beta < 1$, $\varepsilon'_\beta(V_n, R_n) \xrightarrow[n \to \infty]{P} \varepsilon(M)$, where $\varepsilon(M)$ is the differential privacy level of $M$.[8]

The counts $(v, r)$ also yield a *privacy level estimation* $p^{-1}\left(\frac{v}{r}\right)$. For every $n \in \mathbb{N}$, the lower bound on the privacy level is a lower bound on the privacy estimation with the required confidence interval. Thus, the accuracy determines the lower bound, up to the effect of the statistical correction that decreases as the number of taken guesses increases. Notice that if the accuracy converges to some value $a$, the resulting bounds converge to $p^{-1}(a)$. Hence, we define the efficacy of ORA using the expected accuracy.

**Definition 3.2** (ORA Efficacy). The efficacy of ORA with a pair vector $Z$, a guesser $G$, and number of elements $n \in \mathbb{N}$ with respect to a randomized algorithm $M$ is $E_{M,Z,G,n} := \mathbb{E}\left[\frac{V_n}{R_n}\right]$.

We show that asymptotic tightness can be characterized as optimal asymptotic efficacy.

**Lemma 3.3** (ORA Asymptotic Tightness and Efficacy). *ORA is asymptotically tight for a randomized algorithm $M : X^* \to \mathcal{O}$ if and only if there exists a sequence of adversary strategies $\{(Z_n, G_n)\}_{n \in \mathbb{N}}$ with unlimited guesses such that $E_{M,Z,G,n} \xrightarrow{n \to \infty} p(\varepsilon(M))$.*

# 4 The Gaps

In this section, we show that ORA is not asymptotically tight for certain algorithms; that is, even with an optimal adversary, the bounds it yields do not approach the algorithm's true privacy level. This is in contrast to classic auditing (Algorithm 2), which is asymptotically tight for all algorithms (Lemma A.18).

We first consider local algorithms, which, as we will see, are amenable to ORA by analogy to classic auditing. Local algorithms operate at the element level without aggregating different elements.

**Definition 4.1** (Local Algorithm). An algorithm $M : X^n \to \mathcal{O}$ is local if there exists a sub-algorithm $M' : X \to \mathcal{O}'$ such that for every $D \in X^n$, $M(D) = (M'(D_1), ..., M'(D_n))$.

Steinke et al. [1] prove asymptotic tightness of ORA for one $\varepsilon$-DP algorithm (see Proposition B.1):

**Definition 4.2** (Local Randomized Response (LRR) [14]). $LRR_\varepsilon : \{-1, 1\}^* \to \{-1, 1\}^*$ is a local algorithm whose sub-algorithm is randomized response: $RR_\varepsilon(x) = \begin{cases} x & \text{w.p. } p(\varepsilon) \\ -x & \text{w.p. } 1 - p(\varepsilon) \end{cases}$.

When ORA is not asymptotically tight, the gap results from three key differences between the threat model underlying the definition of differential privacy and the ORA setting. For each difference, we give an example of an algorithm that is not differentially private (i.e., $\varepsilon(M) = \infty$) and therefore can be tightly audited only if there exists an adversary whose efficacy approaches the perfect efficacy of 1. However, we show that for these algorithms, even with an optimal adversary, the efficacy of ORA is close to the efficacy of random guessing. More details appear in Appendix B.

---

[7]The notation "$\xrightarrow[n \to \infty]{P}$" refers to convergence in probability. For a random variable $A$, we say that $A$ converges in probability to $\infty$ and denote $A \xrightarrow[n \to \infty]{P} \infty$, if for every $M \in \mathbb{R}$, $Pr[A > M] \xrightarrow{n \to \infty} 1$.

[8]The definition requires only the existence of an adversary strategy for which the condition holds. We stress that the adversary strategy may depend on the algorithm, and even on its privacy level. We use this definition because we reason about the inherent limitations of ORA, even with the most powerful adversary.

**(1) Non-worst-case privacy for elements** (Proposition B.3) The differential privacy level is determined by the worst-case privacy loss of any database element. However, in order to guess frequently enough, ORA may need to issue guesses on elements that experience non-worst-case privacy loss with respect to the current output. Consider the Name and Shame algorithm ($NAS$) [15], which randomly selects an element from its input dataset and outputs it. The optimal efficacy of ORA with respect to $NAS$ approaches $1/2$ when its number of guesses must approach infinity, since the auditor will need to issue guesses on many elements about which it received no information.

**(2) Non-worst-case outputs** (Proposition B.6) Differential privacy considers worst-case outputs, whereas in ORA the algorithm is run only once, and the resulting output may not be worst-case in terms of privacy. Consider the All or Nothing algorithm ($AON_p$), which outputs its entire input with some probability $p$ and otherwise outputs null. The optimal efficacy of ORA with respect to $AON_p$ is $\frac{1}{2} + \frac{p}{2}$. If $p$ is small, the probability of "bad" events (in terms of privacy) is low, and hence the efficacy gap of ORA with respect to $AON_p$ is large. (Notice that this issue is inevitable in any auditing method that runs the audited algorithm a limited number of times.)

**(3) Interference** (Proposition B.9) If an algorithm's output aggregates across multiple inputs, there is interference between their effects, and any of them can be guessed well only using knowledge about the others. Differential privacy protects against an adversary that has full knowledge of all inputs except one, whereas in ORA the adversary may have little or no such information. Consider the XOR algorithm, which takes binary input and outputs the XOR of the input bits. For every $n \geq 2$, the optimal efficacy of ORA with respect to $XOR$ is $\frac{1}{2}$. This uncertainty of the adversary is inherent to ORA, since the auditor first samples a database, and this sampling adds a layer of uncertainty.

To summarize: *DP bounds the privacy loss of **every element** from **any output** against an adversary with **full knowledge**. In contrast, ORA captures a more relaxed privacy notion that only protects the **average element** from the **typical output** against an adversary with **partial knowledge**.*

## 5 Efficacy and Asymptotic Tightness of ORA

In this section, we formally show that the gaps described in Section 4 bound the efficacy of ORA. We characterize the optimal efficacy of ORA and the conditions for the asymptotic tightness of ORA.

ORA is a Markovian process: The auditor samples a vector of bits $S \sim U^n := \text{Uniform}\left(\{-1, 1\}^n\right)$. It then uses the fixed pair vector $Z$ and $S$ to define the dataset $D = \mathcal{Z}(S)$, where $\mathcal{Z}$ is the mapping from bit vectors to datasets that the pair vector $Z$ induces. Next, it uses the algorithm $M$ and $D$ to get the output $O \sim M(D)$. Finally, it uses the guesser $G$ and $O$ to obtain the guess vector $T = G(O)$.

For each pair of elements in the pair vector $Z$, guessing which element was sampled is a Bayesian hypothesis testing problem, where there are two possible elements with equal prior probabilities, and each induces an output distribution. The guesser receives an output and guesses from which distribution it was sampled. By linearity of expectation, the efficacy is determined by the mean success rate of the elements, so guessers that are optimal at the element level have optimal efficacy.

When considering guessers that do not abstain from guessing, since the prior is uniform, *maximum likelihood guessers* are optimal. However, in ORA guessers are allowed to abstain from guessing, so we also consider guessers that guess only if the likelihood crosses some threshold. We analyze the efficacy of these guessers using the *distributional privacy loss*, which is the log-likelihood ratio between the output distributions that the different elements induce. It measures the extent to which an output $o$ distinguishes between elements, extending the notion of the privacy loss [16] to account for uncertainty about the dataset.

In the general setting of a randomized algorithm $M : X^n \to \mathcal{O}$ and a product distribution $\Theta = \Theta_1 \times \ldots \times \Theta_n$ over its domain $X^n$, the *distributional privacy loss* with respect to an index $i \in [n]$ and elements $x, y \in X$ is $\ell_{M,\Theta,i,x,y}(o) := \ln\left(\frac{\Pr_{D \sim \Theta, O \sim M(D)}[O=o|D_i=x]}{\Pr_{D \sim \Theta, O \sim M(D)}[O=o|D_i=y]}\right)$. In the context of ORA, we consider the algorithm $M_Z := M \circ \mathcal{Z} : \{-1, 1\}^n \to \mathcal{O}$ that takes the sampled vector $S$ as input, selects elements based on $Z$, and runs $M$ on the resulting dataset $D$ ($M \circ \mathcal{Z}$ is a restriction of $M$ and hence $\varepsilon(M_Z) \leq \varepsilon(M)$), and the uniform distribution $U^n$. We identify the distributional privacy loss as the quantity that captures all sources of uncertainty in the ORA process, and use

the shorthand $\ell_{M,Z,i}(o) := \ell_{M\circ\mathcal{Z},U^n,i,-1,1}(o) = \ln\left(\frac{\Pr_{S\sim U^n, O\sim M(\mathcal{Z}(S))}[O=o|S_i=-1]}{\Pr_{S\sim U^n, O\sim M(\mathcal{Z}(S))}[O=o|S_i=1]}\right)$. Maximum likelihood guessers first set a threshold $\tau$ which might depend on $o$ and make a decision only for indexes where $|\ell_{M,Z,i}(o)| \geq \tau$, in which case $T_i = \text{sign}(\ell_{M,Z,i}(o))$.[9]

We show that the optimal efficacy of ORA can be characterized using a series of relaxations of differential privacy that capture the efficacy gaps we discuss in Section 4: *distributional differential privacy (DDP)* (based on noiseless privacy [17]), $\varepsilon_D(M,\Theta) := \sup_{o\in\mathcal{O}, i\in[n], x,y\in X} \ell_{M,\Theta,i,x,y}(o)$,[10]

*average-case distributional differential privacy (AC-DDP)*,

$$\varepsilon_{\text{AC}}(M,\Theta) := \mathbb{E}_{O\sim M(\Theta)}\left[\sup_{i\in[n], x,y\in X} p\left(|\ell_{M,\Theta,i,x,y}(O)|\right)\right],[11]$$

and *average-element average-case distributional differential privacy (AE-AC-DDP)*,

$$\varepsilon_{\text{AE-AC}}(M,\Theta) := \mathbb{E}_{O\sim M(\Theta)}\left[\frac{1}{n}\sum_{i=1}^{n}\sup_{x,y\in X} p\left(|\ell_{M,\Theta,i,x,y}(O)|\right)\right].$$

DDP is a relaxation of DP to the case where the adversary knows only the distribution from which the data is sampled, and is closely related to the interference gap. AC-DDP further relaxes DDP by averaging over outputs; it relates to the non-worst-case outputs gap. AE-AC-DDP relaxes further by averaging over elements; it relates to the non-worst-case privacy for elements gap. Notice that in ORA, due to the binary domain $X = \{-1,1\}$ and the anti-symmetry of the privacy loss, these definitions can be simplified by considering $\ell_{M,Z,i}$ without taking the supremum over $x,y\in X$.

## 5.1 Efficacy

We first consider ORA without abstentions; that is, the guesser must guess for every element. In this setting, the guesser's success rate is determined by the average distinguishability of an element in the presence of uncertainty about the other elements, so AE-AC-DDP captures the optimal efficacy.

**Theorem 5.1** (Optimal Efficacy Without Abstentions). *For every algorithm $M$ and a pair vector $Z$,*

$$E_{M,Z,n}^* = \varepsilon_{\text{AE-AC}}(M_Z, U^n) \overset{(1)}{\leq} \varepsilon_{\text{AC}}(M_Z, U^n) \overset{(2)}{\leq} p\left(\varepsilon_D(M_Z, U^n)\right) \overset{(3)}{\leq} p\left(\varepsilon(M_Z)\right),$$

*where $E_{M,Z,n}^*$ denotes the efficacy of the maximum likelihood guesser that takes all $n$ guesses.*

This shows that the gap between AE-AC-DDP and DP is precisely the combination of the three gaps discussed in the previous section. Each inequality in the theorem statement corresponds to one of them: (1) corresponds to the non-worst-case privacy for elements gap, (2) to the non-worst-case outputs gap, and (3) to the interference gap.

In Appendix C.1.1, we show that the optimal efficacy can be characterized using total-variation, in contrast to the max-divergence that characterizes DP (see Proposition C.4).

Abstentions allow the guesser to make only high-confidence guesses, rather than having to guess about every example, increasing the efficacy. There is a tradeoff between the number of guesses and the efficacy: issuing only the highest-confidence guesses increases efficacy but decreases the statistical significance. To handle this tradeoff, we consider guessers that commit to guess at least $k$ guesses for some $k\in[n]$, and denote the optimal efficacy under this constraint by $E_{M,Z,n,k}^*$. In this case, the maximum likelihood guesser that first sorts the distributional privacy losses by absolute value $|\ell_{M,Z,i}(o)|$ and sets $\tau$ to be the $k$th largest is optimal.

We define a privacy notion which is similar to AE-AC-DDP, but averages only over the $k$ elements with the highest absolute value of the distributional privacy loss,

$$\varepsilon_{\text{AE-AC}}^k(M,\Theta) := \mathbb{E}_{O\sim M(\Theta)}\left[\frac{1}{k}\sum_{i\in I_k(O)}\sup_{x,y\in X} p\left(|\ell_{M,\Theta,i,x,y}(O)|\right)\right],$$

---

[9] For simplicity, we assume the loss is computationally feasible, though this might be a source of an additional gap in all auditing methods.

[10] This is the definition for the discrete case. In the continuous case, we use the max-divergence.

[11] $p(x) := \frac{e^x}{e^x+1}$, as defined in Section 2.

where $I_k(o)$ is the set of $k$ indices with the highest absolute value of the distributional privacy loss for an output $o$.

**Theorem 5.2** (Optimal Efficacy)**.** *For every algorithm $M$ and a pair vector $Z$,*

$$E_{M,Z,n,k}^* = \varepsilon_{AE\text{-}AC}^k(M_Z, U^n) \overset{(1)}{\leq} \varepsilon_{AC}(M_Z, U^n) \overset{(2)}{\leq} p\left(\varepsilon_D(M_Z, U^n)\right) \overset{(3)}{\leq} p\left(\varepsilon(M_Z)\right),$$

This result formalizes the importance of abstentions—they mitigate the non-worst-case privacy for elements gap, as they allow the efficacy to be determined only by the $k$ highest privacy losses.

We use these results to analyze ORA of key families of algorithms and calculate the optimal efficacy for concrete algorithms. In Appendix C.5, we analyze local algorithms and focus on the classic Laplace noise addition mechanism, showing a significant auditing gap for ORA due to the non-worst-case outputs gap (see Theorem C.13 and Figure 4). In Appendix D.2.1, we analyze symmetric algorithms and focus on counting queries, showing that the efficacy of ORA for such queries approaches the efficacy of random guessing, due to a combination of the non-worst-case outputs gap and the interference gap (see Proposition D.2).

Figures 1 and 5 empirically demonstrate the effect of the ratio of issued guesses $\frac{k}{n}$ on the auditing results of the DP-SGD algorithm. The experimental setting is described in Section 6.2. As the number of issued guesses increases, the efficacy and the closely related estimation of $\varepsilon$ (i.e., $p^{-1}\left(\frac{v}{r}\right)$) decrease, as expected from Theorem 5.2, since the guesser is forced to guess in cases where it is less confident. However, the statistically corrected bounds on $\varepsilon$ illustrate the tradeoff: issuing less-confident guesses decreases the efficacy but increases the statistical power.

## 5.2 Asymptotic Tightness

Next we use the efficacy characterization and Lemma 3.3 to characterize the conditions for asymptotic tightness.

In Appendix C.3 we focus on ORA without abstentions and characterize the algorithms for which there is no efficacy gap and ORA is asymptotically tight. We show these are the algorithms that can be post-processed to an algorithm that approaches Local Randomized Response (LRR) with their privacy level, where by "approaches LRR" we mean that w.h.p. (over $Z$ and $M$) the post-processed output's distribution is close to Randomized Response for nearly all elements (see Theorem C.10).

When the guesser is allowed to abstain, it suffices that enough elements, for example a constant fraction of them, are sufficiently exposed, and hence the guesser can accurately guess them. We show that ORA is asymptotically tight for an algorithm if and only if the number of elements whose distributional privacy loss is close to $\varepsilon(M)$ is unlimited.

**Theorem 5.3** (Condition for Asymptotic Tightness of ORA)**.** *ORA is asymptotically tight for a randomized algorithm $M : X^* \to \mathcal{O}$ and sequence of pair vectors $\{Z_n \in X^{2n}\}_{n \in \mathbb{N}}$ if and only if for every $\varepsilon' < \varepsilon(M)$,*

$$\left|\{i \in [n] : |\ell_{M,Z_n,i}| \geq \varepsilon'\}\right| \xrightarrow[n \to \infty]{P} \infty.$$

As a corollary, ORA is asymptotically tight for all local algorithms (Definition 4.1). This also follows from the asymptotic tightness of classic auditing (Algorithm 2 and Lemma A.18), by the observation that one-run auditing of a local algorithm can simulate classic auditing of the sub-algorithm.

**Corollary 5.4** (ORA is Asymptotically Tight for Local Algorithms)**.** *ORA is asymptotically tight with respect to every local randomized algorithm $M : X^* \to \mathcal{O}$.*

Local algorithms process each element separately, eliminating the concern of interference, and thus the distributional privacy loss equals the privacy loss of the sub-algorithm. Even if the sub-algorithm has non-worst-case outputs when the number of elements increases, the number of elements that experience worst-case outputs is unlimited. Hence, local algorithms do not suffer from the gaps, and ORA is asymptotically tight for them.

We extend the analysis to partially-local algorithms operating separately on subsets of elements in Appendix D.2.2, where we use Theorem 5.3 to relate the auditing tightness to the algorithm's "degree of locality" (Theorem D.5). The next section discusses a special case of this setting in the context of DP-SGD.

# 6 DP-SGD: A Case Study of ORA and Mitigating Interference

In this section, we step beyond general characterization theorems to consider how well the workhorse algorithm of private learning, DP-SGD, can be audited in one run. DP-SGD is presented in Appendix D.1; it differs from traditional SGD by adding noise to the gradients computed at each update after clipping their norm. We use it as a case study to illustrate our theoretical insights, focusing on the interference gap (introduced in Section 4 and further explored in Section 5), and propose approaches to mitigate the gap.

We audit the DP-SGD algorithm in the *white box access with gradient canaries threat model*. This is the strongest setting that Steinke et al. [1] consider, and thus the most interesting for revealing ORA's limitations. In this threat model, the adversary can insert arbitrary auditing gradients into the training process and observe all intermediate models. Since using some "real" training examples alongside the auditing examples would only decrease the performance of the auditing, we consider ORA where all the examples are auditing examples, matching our definition of ORA. The auditing elements are $n$ $d$-dimensional vectors, each included or not according to $S$ (equivalently, one element of each pair in a pair vector is the zero vector). Each update step of DP-SGD is a noisy multi-dimensional sum of a random batch of the auditing "gradients."

The multi-dimensional sum aggregates the gradients, creating interference between them, which raises a concern about how effectively ORA can audit DP-SGD. The *Dirac canary* attack, first introduced by Nasr et al. [7], is one possible response to this concern. The attack is an approach to ORA that limits the number of auditing gradients to the dimension of the model, and sets each gradient to be zero in all indices except its distinct coordinate, which it sets to the clipping radius. While the interference between random gradients is already small in high dimensions, the Dirac canary approach completely eliminates it, and with high enough dimension, it can allow for meaningful ORA of DP-SGD.

Prior to our work, it seems that the possibility that including multiple elements per coordinate could be beneficial to ORA was overlooked. However, limiting the number of auditing elements to the dimension of the model comes at a cost: it limits the number of elements experiencing high privacy loss and thus the number of high-confidence guesses. This effect is especially severe if the dimension is low or the probability of worst-case outputs is low. Our theoretical and empirical analyses below show that **assigning more than one element per coordinate can outperform ORA with only one element per coordinate**, by optimizing the tradeoff between the benefit of more potentially accurate guesses and the added interference.

To further mitigate the effects of interference when assigning multiple elements per coordinate, in Section 6.3 we propose **Adaptive ORA** (AORA), a new variant of ORA in which the guesser is allowed to use the true value of the sampled bits from $S$ that it has already guessed to better guess the values of subsequent elements. We also show that this adaptivity has benefits, both theoretically and empirically.

## 6.1 Theoretical Analysis

The adversary observes the intermediate model weights and knows the learning rate, so it can compute the noisy gradient sum at each step. Since we take each gradient to be non-negative in a single coordinate, auditing DP-SGD is equivalent to auditing a multi-iteration noisy version of an algorithm we call Count-In-Sets, on random batches of the inputs. The Count-In-Sets algorithm releases multiple counting queries of disjoint subsets of size $s(n)$ of its input (in the case of DP-SGD, the sets are composed of identical gradients).

**Definition 6.1** (Count-In-Sets). The Count-in-Sets algorithm $CIS_s^n : \{0,1\}^n \to \{0,...,s\}^{\lceil \frac{n}{s} \rceil}$ groups its input elements by their order into sets of size (at most) $s(n)$ and outputs the number of ones in each set.

While the noise addition and the sampling of the batches are the randomness sources that make DP-SGD differentially private, the aggregation of the gradients does not contribute to formal privacy guarantees but limits the efficacy of ORA. Hence the Count-In-Sets algorithm distills the interference gap in DP-SGD for theoretical analysis.

In the case of $s(n) = 1$, which corresponds to one element per coordinate, Count-In-Sets simply outputs each input element; ORA, even without abstentions, audits this algorithm tightly. At the other

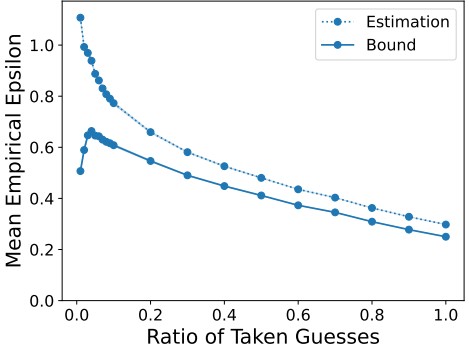
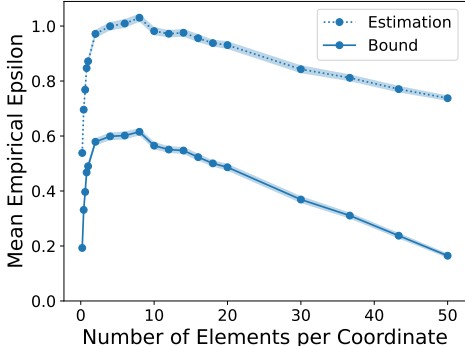

Figure 1: Effect of the fraction of taken guesses $\frac{k}{n}$ on ORA's results for $n = 5000$ elements. While taking only the best guesses increases the privacy estimations, the statistically corrected bounds experience a tradeoff.

Figure 2: Effect of the number of elements per coordinate $\frac{n}{d}$ on ORA's results when making $k = 100$ guesses. The increased number of elements creates a tradeoff, and it is optimized with more than one element per coordinate.

extreme, when $s(n) = n$, we get a single counting query, so the interference is maximal and the optimal efficacy of ORA approaches the minimal $\frac{1}{2}$ efficacy; in particular, ORA is not asymptotically tight (see Proposition D.2). However, we notice that effective auditing is possible with $s(n) > 1$, which corresponds to multiple elements per coordinate. If $s(n) = c$ for some $c \in \mathbb{N}$, it is a local algorithm and hence ORA is asymptotically tight for it. In Proposition D.5, we extend this observation and show that ORA is tight so long as $s(n) = o(\log(n))$, showing that one-run auditing is possible with multiple elements per coordinate, and potentially benefits from the additional elements.

## 6.2 Experiments

Our experiments are designed to empirically illustrate the theoretical insights and provide additional intuition, not to serve as a comprehensive evaluation in realistic settings. We audit DP-SGD with dimension $d = 1000$ for $T = 100$ steps, with sample rate $\frac{1}{10}$. Since the adversary knows the parameters, the values of the clipping threshold and the learning rate do not affect the auditing. We fix $\varepsilon = 2$ and $\delta = 10^{-5}$, and use an RDP accountant to compute the noise scale.[12] We set the auditing gradients as described above: each of them is nonzero in exactly one index. When $n \leq d$ there is no overlap between them, and when $n > d$ we choose an equal number of elements for each index. Our guesser sorts the elements by the value of the update's gradient at the coordinate in which the auditing gradient is non-zero; when committed to taking $k$ guesses, it guesses 1 for the highest $\frac{k}{2}$ elements, and $-1$ for the lowest $\frac{k}{2}$ elements.

We plot the bounds on the privacy level that the auditing method outputs; i.e., $\varepsilon'_\beta(r, v)$ with $\beta = 0.05$ corresponding to a confidence level of $95\%$, as "Bound," and the estimations of the privacy level without statistical correction, i.e., $p^{-1}\left(\frac{v}{r}\right)$, as "Estimation".[13] In Figures 1 and 2, each point is the mean of 200 experiments, and the shaded area represents the standard error of the mean.[14]

In Figure 2, we evaluate the effect of the number of elements per coordinate on the auditing results. Steinke et al. [1] experiment with one element per coordinate, and observe that as the number of elements increases, their auditing results improve. We extend this result and show that assigning multiple elements to each coordinate can further improve auditing. For example, with 1 element per coordinate the mean bound on $\varepsilon$ is $0.49(\pm 0.01)$, and with 8 elements per coordinate it is $0.62(\pm 0.01)$. As predicted by our theoretical results, after a certain point, the statistical benefits of more elements (and hence more guesses) are outweighed by the interference.

---

[12]Since the DP-SGD algorithm is $(\varepsilon, \delta)$-DP, we use the $(\varepsilon, \delta)$-DP variant of ORA. We choose a small value of $\delta$ and hence the results are essentially identical to those for $\epsilon$-DP ORA.

[13]Figures 5 and 6 show similar behavior of the empirical efficacy (mean accuracy) in the same experiments.

[14]In some cases, the error is so small that this area is not visible.

### 6.3 Adaptive ORA

We next consider the new AORA method (see Definition E.3), which is identical to ORA except that it lets the guesser know the true values of the previously guessed elements. In Appendix E, we further discuss this method and prove that it is valid despite the additional information the guesser receives (Proposition E.5).

This adaptivity can significantly improve auditing, as we show in Section E.3. For some algorithms, the improvement is maximal—AORA tightly audits them while ORA completely fails, and we demonstrate this with a variant of the XOR algorithm (Section E.3.1). This adaptivity substantially improves auditing of the Count-In-Sets algorithm—it is not only immune to the interference with multiple elements per coordinate, but also allows taking advantage of the increased number of potential guesses, as we show both theoretically and empirically (Section E.3.2).

We compare ORA and AORA of DP-SGD and show that the improvement from adaptivity is also significant for this important algorithm. For simplicity, we experiment with a single-step, full-batch ($T = 1$ and

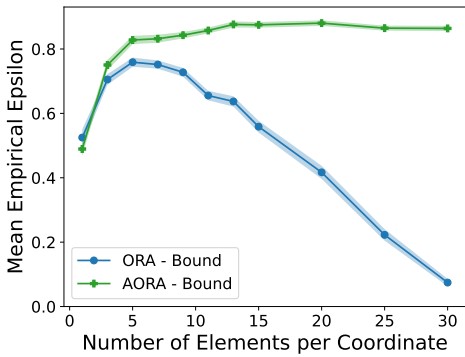

Figure 3: Comparison of the effect of the number of elements per coordinate $\frac{n}{d}$ on the results of ORA and AORA of DP-SGD. AORA outperforms ORA thanks to its resilience to increased interference.

sample rate is 1) version of DP-SGD which can be seen as a noisy version of Count-In-Sets (we leave a comprehensive exploration of auditing of DP-SGD for future work). We set $d = 1000$, $\varepsilon = 2$, and $\delta = 10^{-5}$, as in the previous experiments. To allow for consistent comparison, the guessers for both methods are maximum likelihood guessers with a pre-defined threshold on the privacy loss $\tau = 1$ (unlike the previous experiments in which we fixed the number of guesses). The non-adaptive guesser uses the distributional privacy loss and the adaptive guesser extends it by conditioning on the sampled bits revealed so far.

Figure 3 compares the mean bounds obtained by the two methods. Each point is the mean of 200 experiments. The bounds that AORA produces are higher than those obtained by ORA, thanks to the additional information of the adaptive guesser that allows it to make more high-confidence guesses. Moreover, while the bounds that ORA produces decrease with more than 5 elements per coordinate, AORA only benefits from adding elements. As the effect of interference increases with the number of elements, the non-adaptive guesser has fewer high-confidence guesses to issue. The adaptive guesser may similarly need to abstain from guessing the first elements, but it collects information about these elements that eliminates the effect of the additional interference, allowing it to issue high-confident guesses of later elements. These results confirm that the trends from our simplistic Count-In-Sets analysis (see Section E.3.2 and Figure 9) remain similar in a more complex setting, with noise addition and using a finite threshold on the privacy loss.

## 7 Conclusions

This work characterizes the capabilities of one-run privacy auditing and shows that it faces fundamental gaps. We formalize the three sources of these gaps: non-worst-case privacy for elements, non-worst-case outputs, and interference between elements. These insights clarify the use cases for efficient auditing and can lead to more informed interpretation of auditing results, preventing auditing from being used as a fig leaf for algorithms with poor privacy guarantees. We also introduce new approaches of auditing multiple elements per coordinate and Adaptive ORA to improve auditing by mitigating the effect of interference. Future work can include designing adaptive attacks to leverage the potential of AORA in realistic settings, and exploring the trade-off between the efficiency and effectiveness of privacy auditing, seeking new methodological approaches to optimize it.

## Acknowledgments

This work was supported in part by a gift to the McCourt School of Public Policy and Georgetown University, Simons Foundation Collaboration 733792, Israel Science Foundation (ISF) grant 2861/20, a gift from Apple, and ERC grant 101125913. In addition, Keinan was supported in part by the Federmann Cyber Security Center, and Shenfeld was supported in part by an Apple Scholars in AIML fellowship. Views and opinions expressed are however those of the author(s) only and do not necessarily reflect those of the European Union or the European Research Council Executive Agency. Neither the European Union nor the granting authority can be held responsible for them.

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

# A  Problem Setting

In this section, we generalize the ideas and definitions presented in Section 3 for ORA to a wider family of auditing methods, which we call guessing-based auditing methods. We generalize Lemma 3.3 and prove it. In addition, we present classic auditing and show it is valid and asymptotically tight.

## A.1  Guessing-Based Auditing Methods

*Guessing-based auditing methods* bound the privacy level of a given algorithm according to the success of an adversary in a guessing game. Such auditing methods are defined by an auditor $\mathcal{A}$ that gets oracle access to an algorithm $M : X^n \to \mathcal{O}$ to audit, and takes as input an adversary strategy $f$, which defines how the adversary selects datasets and how it makes guesses based on the algorithm's outputs, and a number of potential guesses $c \in \mathbb{N}$ (if it is not determined by the size of input dataset $n$).

---

**Algorithm 1** One-Run Auditor

1: **Input:** algorithm $M : X^n \to \mathcal{O}$, pair vector $Z = (x_1, y_1, ..., x_n, y_n) \in X^{2n}$ such that for all $i \in [n]$, $x_i \neq y_i$ , guesser $G : \mathcal{O} \to \{-1, 0, 1\}^n$.
2: **for** $i = 1$ to $n$ **do**
3:     Sample $S_i \in \{-1, +1\}$ uniformly.
4: **end for**
5: Define a dataset $D \in X^n$ by $D_i = \begin{cases} x_i & \text{if } S_i = -1 \\ y_i & \text{if } S_i = 1 \end{cases}$ .
6: Compute $o = M(D)$.
7: Guess $T = G(o) \in \{-1, 0, 1\}^n$.
8: Count the numbers of correct guesses $v := |\{i \in [n] : T_i = S_i\}|$ and taken guesses $r := |\{i \in [n] : T_i \neq 0\}|$.
9: **Return:** $v, r$

---

**Algorithm 2** Classic Auditor

1: **Input:** randomized algorithm $M : X^n \to \mathcal{O}$, dataset $D_{\text{base}} \in X^n$, index $j \in [n]$, elements $x, y \in X$, guesser $G : \mathcal{O} \to \{-1, 0, 1\}$, number of potential guesses $c$.
2: **for** $i = 1$ to $c$ **do**
3:     Sample $S_i \in \{-1, +1\}$ uniformly.
4:     Define $d = \begin{cases} x & \text{if } S_i = -1 \\ y & \text{if } S_i = 1 \end{cases}$ and define $D$ as the resulting dataset from replacing the $j$'th element of the base dataset $D_{\text{base}}$ with the element $d$.
5:     Compute $o = M(D)$.
6:     Guess $T_i = G(o) \in \{-1, 0, 1\}$.
7: **end for**
8: Count the numbers of taken guesses $r := |\{i \in [c] : T_i \neq 0\}|$ and accurate guesses $v := |\{i \in [c] : T_i = S_i\}|$.
9: **Return:** $r, v$

---

Both classic auditing and one-run auditing are guessing-based auditing methods. In classic auditing, a variant of the auditing method that Jagielski et al. [6] introduce (Algorithm 2), an adversary strategy $f$ is a base dataset $D_{\text{base}} \in X^n$, an index $i \in [n]$, a pair of elements $x \neq y \in X$, and a guesser $G$. In ORA, a strategy $f$ is a pair vector $Z = (x_1, y_1, ..., x_n, y_n) \in X^{2n}$ and a guesser $G$. In classic auditing, the number of potential guesses $c$ is an input to the auditor, whereas in ORA $c = n$.

In these auditing methods, similarly to the description of ORA in Section 2, The auditor $\mathcal{A}$ samples a random vector $S \in \{-1, 1\}^c$ uniformly, defines datasets according to $S$ and the adversary strategy $f$, and feeds the resulting data to $M$ to get outputs. The auditor then uses the outputs and $f$ to produce a vector of guesses $T \in \{-1, 0, 1\}^c$ for the random bits of $S$. It then computes the number of correct guesses $v$ and the number of taken guesses $r$. When the adversary strategy $f$ is clear from the context, we denote the corresponding random variables by $V_c$ and $R_c$. These counts yield a privacy level estimation $p^{-1}\left(\frac{v}{r}\right)$ and a corresponding lower bound $\varepsilon'_\beta(v, r)$, which are defined as for ORA.

We say that an auditing method is valid if its outputs always yield lower bounds on the privacy level of the audited algorithm with the required confidence level.

**Definition A.1** (Validity). An auditor is valid if for every randomized algorithm $M : X^n \to \mathcal{O}$, adversary strategy $\{f_c\}_{c \in \mathbb{N}}$, $\beta \in (0, 1)$, and $c \in \mathbb{N}$,

$$Pr[\varepsilon'_\beta(V_c, R_c) \leq \varepsilon(M)] \geq 1 - \beta.$$

If the number of accurate guesses $V_c$ is stochastically dominated (see Definition A.6) by the Binomial distribution with $R_c$ trials and success probability of $p\,(\varepsilon(M))$, the auditing method is valid (see Lemma A.7). Both Classic Auditing and ORA are proven to be valid (see Proposition A.8 and Proposition A.9).

If the number of guesses that $\mathcal{A}$ takes approaches $\infty$ as $c$ increases, the lower bounds converge to the privacy estimations. Hence, to analyze the asymptotic behavior of the bounds we can ignore the statistical correction effect and focus on the asymptotic behavior of the privacy level estimations (see Lemma A.13 for a formal treatment). We say that an auditing method is asymptotically valid if its privacy estimations asymptotically lower bound the privacy level of the algorithm.

**Definition A.2** (Asymptotic Validity). An auditor $\mathcal{A}$ is asymptotically valid if for every randomized algorithm $M : X^* \to \mathcal{O}$ and adversary strategy $\{f_c\}_{c \in \mathbb{N}}$, for every $a > 0$,

$$Pr\left[p^{-1}\left(\frac{V_c}{R_c}\right) \leq \varepsilon(M) + a\right] \xrightarrow{c \to \infty} 1.$$

Validity implies asymptotic validity (see Proposition A.14).

We say that an auditor is asymptotically tight for an algorithm if there exists an adversary strategy such that the number of guesses it takes approaches $\infty$ and its estimations asymptotically upper bound the privacy level of the algorithm.

**Definition A.3** (Asymptotic Tightness). An auditor $\mathcal{A}$ is asymptotically tight for a randomized algorithm $M : X^* \to \mathcal{O}$ if there exists an adversary strategy $\{f_c\}_{c \in \mathbb{N}}$ with unlimited guesses, i.e., $R_c \xrightarrow[c \to \infty]{P} \infty$, such that for every $a > 0$,

$$Pr\left[p^{-1}\left(\frac{V_c}{R_c}\right) \geq \varepsilon(M) - a\right] \xrightarrow{c \to \infty} 1.$$

Since the asymptotic behavior of the privacy estimations and the privacy bounds is identical, for valid auditors, asymptotic tightness is equivalent to the requirement that for a large enough number of guesses, with high probability, the lower bounds that the corresponding auditing method outputs approach the algorithm's privacy level. Hence, the definition above is consistent with Definition 3.1 (see Proposition A.15 for a formal treatment).

The efficacy of an auditor $\mathcal{A}$ is defined as in Definition 3.2 as the expected accuracy

$$E_{M,f,c} := \mathbb{E}\left[\frac{V_c}{R_c}\right].$$

We extend Lemma 3.3 to general guessing-based auditing methods and prove it in Section A.5.

**Lemma A.4** (Efficacy and Asymptotic Tightness). *For every asymptotically valid auditor $\mathcal{A}$ with unlimited guesses, it is asymptotically tight for a randomized algorithm $M : X^* \to \mathcal{O}$ if and only if there exists an adversary strategy $\{f_c\}_{c \in N}$ such that $E_{M,f_c,c} \xrightarrow{c \to \infty} p\,(\varepsilon(M))$.*

Lemma A.18 uses the above lemma to show that classic auditing is asymptotically tight with respect to every algorithm.

## A.2  Validity

We define the lower bound of the Clopper-Pearson confidence interval. This is the function that converts the number of taken guesses and the number of accurate guesses to a lower bound on the success probability.

**Definition A.5** (Clopper-Pearson Lower Bound). The Clopper-Pearson lower bound $CPL(r, v, 1-\beta)$ of a number of trials $r \in \mathbb{N}$, number of successes $v \in \mathbb{N}$ such that $v \leq r$, and confidence level $1 - \beta \in (0, 1)$ is

$$CPL(r, v, \beta) := sup\{p \in [0, 1] : Pr[\text{Bin}(r, p) \geq v] \leq \beta\}.$$

We show a condition on the distribution of the outputs of the auditor that implies validity of the auditing method. First, we present stochastic dominance, a partial order between random variables.

**Definition A.6** (Stochastic Dominance). A random variable $X \in \mathbb{R}$ is stochastically dominated by a random variable $Y \in \mathbb{R}$ if for every $t \in \mathbb{R}$

$$Pr[X > t] \leq Pr[Y > t].$$

In this case, we denote $X \preccurlyeq Y$.

**Lemma A.7** (Condition for Validity). *For every auditor $\mathcal{A}$, randomized algorithm $M : X^* \to \mathcal{O}$, adversary strategy $f$, $\beta \in (0, 1)$, and number of guesses $c$, if $V_c \preccurlyeq Binomial(R_c, p(\varepsilon(M)))$, then $\mathcal{A}$ is valid.*

*Proof.* Fix a randomized algorithm $M : X^n \to \mathcal{O}$, adversary strategy $f$, $\beta \in (0, 1)$, and $c \in \mathbb{N}$, and denote $V' \sim \text{Bin}(R_c, p(\varepsilon(M)))$. We have that

$$Pr[\varepsilon'_\beta(V_c, R_c) \leq \varepsilon(M)] = Pr[CPL(R_c, V_c) \leq p(\varepsilon(M))] \text{ (By the monotonicity of } p)$$
$$\geq Pr[CPL(R_c, V') \leq p(\varepsilon(M))] \text{ (By stochastic domination and monotonicity of CPL))}$$
$$= 1 - \beta \text{ (By the CPL definition and the continuity of a binomial's CDF in } p).$$

Therefore, $\mathcal{A}$ is valid. $\square$

Differential privacy can be characterized as a requirement on the classic auditing game: for every adversary strategy, the probability to guess correctly when a guess is made is bounded by $p(\varepsilon(M))$. Since in classic auditing, the rounds are independent of each other, the number of accurate guesses is stochastically dominated by $\text{Bin}(r, p(\varepsilon))$. Hence, as Jagielski et al. [6] show for their version of classic auditing, the version we present here is valid; that is, the output of classic auditing yields a lower bound on the privacy level.

**Proposition A.8** (Classic Auditing is Valid). *Classic auditing is a valid guessing-based auditing method.*

*Proof.*

$$Pr[S_i = -1 | T_i = t] = \frac{Pr[T_i = t | S_i = -1] Pr[S_i = -1]}{Pr[T = t]} \text{ (Bayes' law)}$$
$$= \frac{Pr[T_i = t | S_i = -1] \cdot \frac{1}{2}}{Pr[T = t]} \text{ (} S_i \text{ is sampled uniformly)}$$
$$= \frac{Pr[T_i = t | S_i = -1] \cdot \frac{1}{2}}{\frac{1}{2} Pr[T = t | S_i = -1] + \frac{1}{2} Pr[T = t | S_i = 1]} \text{ (Law of total probability)}$$
$$= \frac{Pr[T_i = t | S_i = -1] / Pr[T_i = t | S_i = 1]}{Pr[T_i = t | S_i = -1] / Pr[T_i = t | S_i = 1] + 1} \text{ (algebra)}$$
$$\in \left[ \frac{e^{-\varepsilon(M)}}{e^{-\varepsilon(M)} + 1}, \frac{e^{\varepsilon(M)}}{e^{\varepsilon(M)} + 1} \right]$$
$$= [1 - p(\varepsilon(M)), p(\varepsilon(M))],$$

where the second-to-last line holds because the guess $T_i$ is a post-processing of the output of the $\varepsilon(M)$-differentially private $M$.

Therefore, for every taken guess $T_i \neq 0$, the success probability is bounded by $p(\varepsilon)$. The rounds are independent so the number of accurate guesses is stochastically dominated as $V \preccurlyeq Binomial(R, p(\varepsilon(M)))$. $\square$

Steinke et al. [1] show that the probability of success in every guess is bounded by $p(\varepsilon)$, even if one conditions on the previous sampled bits of $S$. We use their result to formally show that ORA is valid.

**Proposition A.9** (One-Run Auditing is Valid). *One-run auditing is a valid guessing-based auditing method.*

*Proof.* Steinke et al. [1] show that for every randomized algorithm $M : X^n \to \mathcal{O}$ and $t \in \{-1, 0, 1\}^n$, $V|(T = t) \preceq \text{Bin}(\|t\|_1, p(\varepsilon(M)))$. By the law of total probability, $V \preceq \text{Bin}(\|T\|_1, p(\varepsilon(M))) = \text{Bin}(R, p(\varepsilon(M)))$. $\qquad\square$

## A.3 Asymptotic Validity

We show that the Clopper-Pearson bounds converge to the ratio of successful guesses, and that the distance between these quantities is bounded by the number of trials.

**Lemma A.10** (Convergence of Clopper-Pearson Lower). *For every number of trials $r \in \mathbb{N}$, number of successes $v \in \mathbb{N}$ such that $v \leq r$, and confidence level $1 - \beta \in (0, 1)$,*

$$\left| \frac{v}{r} - CPL(r, v, \beta) \right| \leq \sqrt{-\frac{\ln \beta}{2r}}.$$

*Proof.* We denote $V' \sim \text{Bin}(r, CPL(r, v, \beta))$ and use Hoeffding's inequality to bound the probability of $Y$ to be greater than or equal to $v$:

$$
\begin{aligned}
\beta &= Pr[V' \geq v] \\
&= Pr[V' - r \cdot CPL(r, v, \beta) \geq v - r \cdot CPL(r, v, \beta)] \\
&\leq exp\left(-2\frac{(v - r \cdot CPL(r, v, \beta))^2}{r}\right),
\end{aligned}
$$

where the first equality follows from the property of the Clopper-Pearson lower bound and the continuity of the binomial distribution's PMF with respect to the success probability, the second equality follows from algebraic manipulation, and the last inequality is an application of Hoeffding's inequality. Further algebraic manipulation yields

$$\left| \frac{v}{r} - CPL(r, v, \beta) \right| \leq \sqrt{-\frac{\ln \beta}{2r}}. \qquad\square$$

We show that in the setting where the numbers of trials and successes are random variables, if the number of trials approaches $\infty$, the difference between the success ratio and the Clopper-Pearson bounds converges to $0$.

**Lemma A.11** (Convergence of Clopper-Pearson Lower for Random Variables). *Given a probability space $(\Omega, \mathcal{F}, P)$, for every pair of sequences of random variables over $\Omega$, $\{V_c : \Omega \to \mathbb{N}\}_{c \in \mathbb{N}}$ and $\{R_c : \Omega \to \mathbb{N}\}_{c \in \mathbb{N}}$ such that for every $c \in \mathbb{N}$, $V_c \leq R_c$, and confidence level $\beta \in (0, 1)$, if $R_c \xrightarrow[c \to \infty]{P} \infty$, then*

$$\left| \frac{V_c}{R_c} - CPL(R_c, V_c, \beta) \right| \xrightarrow[c \to \infty]{P} 0.$$

*Proof.* Let $\epsilon > 0$ and $\delta > 0$. From Lemma A.10, and since for every $\beta \in (0, 1)$, $\sqrt{-\frac{\ln \beta}{2r}} \xrightarrow{r \to \infty} 0$, there exists $N \in \mathbb{N}$ such that for every $r > N$ and $v \in \mathbb{N}$ such that $v \leq r$,

$$\left| \frac{v}{r} - CPL(r, v, \beta) \right| \leq \epsilon.$$

Since $R_c \xrightarrow[c \to \infty]{P} \infty$, there exists $C \in \mathbb{N}$ such that for every $c > C$,

$$P[R_c > N] \geq 1 - \delta.$$

Hence, for every $c > C$,

$$P\left[ \left| \frac{V_c}{R_c} - CPL(R_c, V_c, \beta) \right| \leq \epsilon \right] \geq 1 - \delta.$$

Therefore,

$$\left| \frac{V_c}{R_c} - CPL(R_c, V_c, \beta) \right| \xrightarrow[c \to \infty]{P} 0. \qquad \square$$

We show that asymptotic validity can be characterized as a condition over the privacy lower bounds rather than the privacy estimations.

**Lemma A.12** (Privacy Estimations and Bounds). *For every auditor $\mathcal{A}$ with unlimited guesses, randomized algorithm $M : X^* \to \mathcal{O}$, adversary strategy $\{f_c\}_{c \in \mathbb{N}}$, and $\beta \in (0,1)$,*

$$p^{-1}\left( \frac{V_c}{R_c} \right) - \varepsilon'_\beta(V_c, R_c) \xrightarrow[c \to \infty]{P} 0.$$

*Proof.*

$$\left| p^{-1}\left( \frac{V_c}{R_c} \right) - \varepsilon'_\beta(V_c, R_c) \right|$$

$$= \left| p^{-1}\left( \frac{V_c}{R_c} \right) - sup(\{a \in \mathbb{R}^+ : Pr[\text{Binomial}(R_c, p\,(a)) \geq V_c] \leq \beta\}) \right| \text{ (by definition)}$$

$$= \left| p^{-1}\left( \frac{V_c}{R_c} \right) - p^{-1}\left( sup(\{q \in \mathbb{R}^+ : Pr[\text{Binomial}(R_c, q) \geq V_c] \leq \beta\}) \right) \right| \text{ (monotonicity of } p^{-1})$$

$$= \left| p^{-1}\left( \frac{V_c}{R_c} \right) - p^{-1}\left( CPL(R_c, V_c, \beta) \right) \right| \text{ (by definition)}$$

$$\xrightarrow[c \to \infty]{P} 0 \text{ (By Lemma A.11 because } \mathcal{A} \text{ has unlimited guesses).}$$

$$\square$$

**Proposition A.13.** *An auditor $\mathcal{A}$ with unlimited guesses is asymptotically valid if and only if for every randomized algorithm $M : X^* \to \mathcal{O}$, adversary strategy $\{f_c\}_{c \in \mathbb{N}}$, and $\beta \in (0,1)$, for every $a > 0$,*

$$Pr\left[ \varepsilon'_\beta(V_c, R_c) \leq \varepsilon(M) + a \right] \xrightarrow{c \to \infty} 1.$$

*Proof.* By Lemma A.12, the difference between the privacy estimations and the privacy bounds approach 0, and thus

$$\forall a > 0 : Pr\left[ \varepsilon'_\beta(V_c, R_c) \leq \varepsilon(M) + a \right] \xrightarrow{c \to \infty} 1$$

$$\iff \forall a > 0 : Pr\left[ p^{-1}\left( \frac{V_c}{R_c} \right) \leq \varepsilon(M) + a \right] \xrightarrow{c \to \infty} 1,$$

and the claim follows. $\qquad \square$

**Proposition A.14** (Validity implies Asymptotic Validity). *Every valid auditor $\mathcal{A}$ is asymptotically valid.*

*Proof.* Let $\mathcal{A}$ be a valid auditor and assume by contradiction that it is not asymptotically valid, that is, there exist an algorithm $M$, an adversary strategy $\{f_c\}_{c \in \mathbb{N}}$, $a > 0$ and $\delta > 0$ such that for infinitely many values of $c$

$$Pr\left[ p^{-1}\left( \frac{V_c}{R_c} \right) > \varepsilon(M) + a \right] > \delta.$$

From Lemma A.11 and the continuity of $p^{-1}$, for every $\beta \in (0,1)$, for sufficiently large $c$,

$$Pr\left[ \varepsilon'_\beta(V_c, R_c) \geq p^{-1}\left( \frac{V_c}{R_c} \right) - a \right] \geq 1 - \frac{\delta}{2}.$$

Hence, for every $\beta > 0$ there exists $c$ such that

$$Pr\left[ \varepsilon'_\beta(V_c, R_c) > \varepsilon(M) \right] > \frac{\delta}{2}.$$

For $\beta = \frac{\delta}{2}$, this is a contradiction to validity, completing the proof. $\qquad \square$

### A.4 Asymptotic Tightness

**Proposition A.15** (Characterization of Asymptotic Tightness of Valid Auditors). *For every valid auditor $\mathcal{A}$, $\mathcal{A}$ is asymptotically tight if and only if there exists a strategy $\{f_c\}_{c\in\mathbb{N}}$ with unlimited guesses, i.e., $R_c \xrightarrow[c\to\infty]{P} \infty$, such that for every confidence level $0 < 1 - \beta < 1$,*

$$\varepsilon'_\beta(V_c, R_c) \xrightarrow[c\to\infty]{P} \varepsilon(M).$$

*Proof.* Let $\{f_c\}_{c\in\mathbb{N}}$ be a strategy. $\mathcal{A}$ is valid, so by Proposition $A.14$ it is asymptotically valid, that is,

$$\forall a > 0 : Pr\left[p^{-1}\left(\frac{V_c}{R_c}\right) \le \varepsilon(M) + a\right] \xrightarrow{c\to\infty} 1.$$

Therefore,

$$\forall a > 0 : Pr\left[p^{-1}\left(\frac{V_c}{R_c}\right) \ge \varepsilon(M) - a\right] \xrightarrow{c\to\infty} 1$$

if and only if

$$p^{-1}\left(\frac{V_c}{R_c}\right) \xrightarrow[c\to\infty]{P} \varepsilon(M).$$

By Lemma A.12, this occurs if and only if

$$\forall \beta \in (0,1) : \varepsilon'_\beta(V_c, R_c) \xrightarrow[c\to\infty]{P} \varepsilon(M),$$

which completes the proof. $\qquad\square$

### A.5 Efficacy and Asymptotic Tightness

**Lemma A.16.** *Let $a, b \in \mathbb{R}$ and $l \in [a,b]$, for every sequence of random variables $\{X_n\}_{n\in\mathbb{N}} \subseteq [a,b]$, if*

$$\forall \epsilon > 0 : Pr[X_n \le l + \epsilon] \xrightarrow{n\to\infty} 1,$$

*then*

$$\forall \epsilon > 0 : Pr[X_n \ge l - \epsilon] \xrightarrow{n\to\infty} 1$$

*if and only if*

$$\mathbb{E}[X_n] \xrightarrow{n\to\infty} l.$$

*Proof.* First, we show that if $\forall \epsilon > 0 : Pr[X_n \ge l - \epsilon] \xrightarrow{n\to\infty} 1$, then $\mathbb{E}[X_n] \xrightarrow{n\to\infty} l$. Using the union bound, for every $\epsilon > 0$, $Pr[l - \epsilon \le X_n \le l + \epsilon] \xrightarrow{n\to\infty} 1$, that is $X_n \xrightarrow[n\to\infty]{P} l$. For bounded random variables, convergence in probability implies convergence in expectation, so $\mathbb{E}[X_n] \xrightarrow{n\to\infty} l$.

It remains to show that if $\mathbb{E}[X_n] \xrightarrow{n\to\infty} l$, then $\forall \epsilon > 0 : Pr[X_n \ge l - \epsilon] \xrightarrow{n\to\infty} 1$. For every $n \in \mathbb{N}$ and $\epsilon > 0$, we denote $P_n^{\mathrm{L}}(\epsilon) := Pr[X_n < l - \epsilon]$ and $P_n^{\mathrm{H}}(\epsilon) := Pr[X_n > l + \epsilon]$ and we have that for every $\epsilon^{\mathrm{L}}, \epsilon^{\mathrm{H}} > 0$,

$$\begin{aligned}
\mathbb{E}[X_n] &\le P_n^{\mathrm{L}}(\epsilon^{\mathrm{L}}) \cdot (l - \epsilon^{\mathrm{L}}) + Pr[l - \epsilon^{\mathrm{L}} \le X_n \le l + \epsilon^{\mathrm{H}}] \cdot (l + \epsilon^{\mathrm{H}}) + P_n^{\mathrm{H}}(\epsilon^{\mathrm{H}}) \cdot b \\
&\le P_n^{\mathrm{L}}(\epsilon^{\mathrm{L}}) \cdot (l - \epsilon^{\mathrm{L}}) + (1 - P_n^{\mathrm{L}}(\epsilon^{\mathrm{L}}) - P_n^{\mathrm{H}}(\epsilon^{\mathrm{H}})) \cdot (l + \epsilon^{\mathrm{H}}) + P_n^{\mathrm{H}}(\epsilon^{\mathrm{H}}) \cdot (l + b - l) \\
&\le l + \epsilon^{\mathrm{H}} - P_n^{\mathrm{L}}(\epsilon^{\mathrm{L}}) \cdot \epsilon^{\mathrm{L}} + P_n^{\mathrm{H}}(\epsilon^{\mathrm{H}}) \cdot (b - l).
\end{aligned}$$

Given $\alpha > 0$ and $\epsilon^{\mathrm{L}} > 0$, take $\epsilon^{\mathrm{H}} = \frac{\alpha\epsilon^{\mathrm{L}}}{3}$. Then for every $n \in \mathbb{N}$,

$$\begin{aligned}
P_n^{\mathrm{L}}(\epsilon^{\mathrm{L}}) &\le \frac{1}{\epsilon^{\mathrm{L}}}(\epsilon^{\mathrm{H}} + l - \mathbb{E}[X_n] + P_n^{\mathrm{H}}(\epsilon^{\mathrm{H}})(b - l)) \\
&= \frac{\alpha\epsilon^{\mathrm{L}}}{3\epsilon^{\mathrm{L}}} + \frac{l - \mathbb{E}[X_n]}{\epsilon^{\mathrm{L}}} + \frac{P_n^{\mathrm{H}}(\frac{\alpha\epsilon^{\mathrm{L}}}{3})(b - l)}{\epsilon^{\mathrm{L}}}.
\end{aligned}$$

By the convergence of expectation, for sufficiently large $n$, $\frac{l-\mathbb{E}[X_n]}{\epsilon^{\mathrm{L}}} \leq \frac{\alpha}{3}$. By the given upper boundedness of $X_n$, for sufficiently large $n$, $\frac{P_n^{\mathrm{H}}(\frac{\alpha\epsilon^{\mathrm{L}}}{3})(b-l)}{\epsilon^{\mathrm{L}}} \leq \frac{\alpha}{3}$. Hence, for sufficiently large $n$,

$$P_n^{\mathrm{L}}(\epsilon^{\mathrm{L}}) \leq \frac{\alpha}{3} + \frac{\alpha}{3} + \frac{\alpha}{3}$$
$$= \alpha.$$

Therefore, for every $\epsilon^{\mathrm{L}} > 0$, $P_n^{\mathrm{L}}(\epsilon^{\mathrm{L}}) \xrightarrow{n\to\infty} 0$, and hence $Pr[X_n \geq l - \epsilon^{\mathrm{L}}] \xrightarrow{n\to\infty} 1$. $\qquad\square$

**Lemma A.17** (Efficacy and Tightness). *(Lemma A.4) For every asymptotically valid auditor $\mathcal{A}$ with unlimited guesses, it is asymptotically tight for a randomized algorithm $M : X^* \to \mathcal{O}$ if and only if there exists an adversary strategy $\{f_c\}_{c\in N}$ such that $E_{M,f_c,c} \xrightarrow{c\to\infty} p(\varepsilon(M))$.*

*Proof.* Let $f$ be an adversary strategy $f := \{f_c\}_{c\in N}$. We show that $A$ is asymptotically tight with $f$ if and only if its efficacy with it converges in probability to $p(\varepsilon(M))$.

Define $\{X_c\}_{c\in\mathbb{N}}$ as the sequence of the random variables of the accuracy for each number of potential guesses, that is, $\frac{V_c}{R_c}$ where $V_c, R_c \sim \mathcal{A}_{f_c,c}(M)$.

We have that $\{X_c\}_{c\in\mathbb{N}} \subseteq [0,1]$, and using the asymptotic validity and the continuity of $p$ we have that

$$\forall a > 0 : Pr[X_c \leq p(\varepsilon(M)) + a] \xrightarrow{c\to\infty} 1.$$

Hence, we can use Lemma A.16 to show that

$$\forall a > 0 : Pr[X_c \geq p(\varepsilon(M)) - a] \xrightarrow{c\to\infty} 1$$

if and only if

$$\mathbb{E}[X_c] \xrightarrow{c\to\infty} p(\varepsilon(M)).$$

Using the continuity of $p$ and $p^{-1}$,

$$\forall a > 0 : Pr[p^{-1}(X_c) \geq \varepsilon(M) - a] \xrightarrow{c\to\infty} 1$$

if and only if

$$\mathbb{E}[X_c] \xrightarrow{c\to\infty} p(\varepsilon(M)).$$

By the definitions of asymptotic tightness and efficacy we get that $A$ is asymptotically tight with $f$ if and only if its efficacy with it converges in probability to $p(\varepsilon(M))$. $\qquad\square$

Classic auditing is asymptotically tight with respect to every algorithm.

**Lemma A.18** (Classic Auditing is Asymptotically Tight). *Classic auditing is asymptotically tight with respect to every randomized algorithm $M : X^n \to \mathcal{O}$.*

*Proof.* By Definition 2.1, there exist $x, y \in X$ such that $D_\infty(M'(x)||M'(y)) = \varepsilon(M)$ and $p := Pr_{O\sim M(X)}[L_{M',x,y} = \varepsilon] > 0$. Consider the guesser $G$ that guesses 1 if $L_{M,x,y} = \varepsilon$ and otherwise abstains from guessing. The rounds are independent, so the number of guesses of $G$ is distributed binomially $V_T \sim \mathrm{Bin}(T, p) \xrightarrow{T\to\infty} \infty$, and hence it has unlimited guesses. By the definition of $G$, its probability to accurately guess a taken guess is $\varepsilon$ so using the weak law of large numbers and the fact it has unlimited guesses, its efficacy converges to $\varepsilon$. Using Lemma A.4, classic auditing is asymptotically tight for $M$. $\qquad\square$

# B  Gaps

**Proposition B.1** (ORA is Asymptotically Tight for Local Randomized Response). *For every $\varepsilon \in [0,\infty]$, ORA without abstentions is asymptotically tight with respect to $\varepsilon$-Local Randomized Response.*

*Proof.* Consider any pair vector $Z$ and the guesser that guesses the output, $G(O) = O$ and never abstains. The guess for the $i$th pair is accurate if and only if $O_i = D_i$, which happens with probability $p(\varepsilon)$ for each element. For every $n \in \mathbb{N}$, the efficacy is $p(\varepsilon)$, so using Lemma A.4, ORA is asymptotically tight for Local Randomized Response. $\qquad\square$

## B.1 Number of Elements Exposed

We use the Name and Shame algorithm [15] to demonstrate the low efficacy of algorithms for which only a limited number of the elements experience high privacy loss.[15]

**Definition B.2** (Name And Shame (NAS) [15]). The Name and Shame algorithm $NAS : X^* \to X$ is the algorithm that randomly selects an element and outputs it.

Name And Shame exposes one of its input elements, and hence is not differentially private.

**Proposition B.3** (ORA Efficacy for NAS). *For every adversary strategy $\{f_n = (Z_n, G_n)\}_{n \in \mathbb{N}}$ with unlimited guesses, the optimal efficacy of ORA with respect to $NAS$ approaches $\frac{1}{2}$; that is,* $E_{NAS,f_n,n} \xrightarrow{n \to \infty} \frac{1}{2}$.

*Proof.* For every pair vector $Z$ and guesser $G$, the success probability in guessing any element except the one that was exposed is $\frac{1}{2}$. If the guesser guesses that the value of the exposed element is the output, its success probability in guessing this element is $1$. Using the law of total probability and the linearity of expectation, and since the adversary has unlimited guesses, the optimal efficacy of ORA with respect to $NAS$ approaches $\frac{1}{2}$. $\square$

We deduce that ORA is not asymptotically tight for NAS.

**Corollary B.4** (ORA is Not Asymptotically Tight for NAS). *ORA is not asymptotically tight for $NAS$.*

*Proof.* Using Lemma A.4, since for every adversary strategy $\{f_n\}_{n \in \mathbb{N}}$, $E_{NAS,f_n,n} \xrightarrow{n \to \infty} \frac{1}{2} < p(\infty) = 1$. $\square$

Name and Shame completely exposes one of its elements, but since ORA requires unlimited guesses (see Lemma A.4), it does not affect the asymptotic efficacy.

## B.2 Non-Worst-Case Outputs

We use the All Or Nothing algorithm to demonstrate how non-worst-case outputs decrease the efficacy of ORA.

**Definition B.5** (All Or Nothing (AON)). The All Or Nothing algorithm $AON_p : X^* \to X^* \cup \{null\}$ is an algorithm parametrized by $p$ that either outputs its input or outputs null.

$$AON_p(D) = \begin{cases} D & \text{with probability } p \\ null & \text{otherwise} \end{cases}$$

$AON_p$ may expose its input, and hence is not differentially private.

**Proposition B.6** (ORA Efficacy for AON). *For every $n \in \mathbb{N}$, $0 < p < 1$, and adversary strategy $(Z, G)$, the optimal efficacy of ORA with respect to $AON_p$ is $\frac{1}{2} + \frac{p}{2}$.*

*Proof.* For every pair vector $Z$, guesser $G$, and element, the success probability when the output $O$ is null is $\frac{1}{2} < 1$. For the guesser that guesses the output $G(O) = O$, the success probability when the output $O$ is not null is $1$. Using the law of total probability and the linearity of expectation, the optimal efficacy of ORA with respect to $AON_p$ is

$$p \cdot 1 + (1 - p) \cdot \frac{1}{2} = \frac{1}{2} + \frac{p}{2}.$$

$\square$

We deduce that ORA is not asymptotically tight for AON.

**Corollary B.7** (ORA is Not Asymptotically Tight for AON). *For every $0 < p < 1$, ORA is not asymptotically tight for $AON_p$.*

---

[15] Aerni et al. [18] use another variation of the algorithm to demonstrate non-optimal usage of privacy estimation methods. We point out that in ORA this gap is inherent to the method.

*Proof.* Using Lemma A.4, since for every adversary strategy $f$ and $n \in \mathbb{N}$, $E_{M,f,n} = \frac{1}{2} + \frac{p}{2} < p(\infty) = 1$. $\square$

If $p$ is small, the probability of "bad" events (in terms of privacy) is low, and hence the efficacy gap of ORA with respect to $AON_p$ is big.

### B.3 Interference

We use the XOR algorithm to illustrate the decrease in efficacy due to the interference between the elements.[16]

**Definition B.8** (XOR). The "XOR" algorithm $XOR : \{0,1\}^* \to \{0,1\}$ is the algorithm that takes binary input and outputs the XOR of the input bits.

$$XOR(D) = D_1 \oplus ... \oplus D_{|D|}.$$

For every $n \in \mathbb{N}$, the $XOR$ algorithm is deterministic and not constant, and hence not differentially private.

For every $n \geq 2$, the optimal efficacy of ORA with respect to $XOR$ is $\frac{1}{2}$, which is the same as in random guessing.

**Proposition B.9** (ORA Efficacy for XOR). *For every $n \geq 2$ and adversary strategy $(Z, G)$, the efficacy of ORA with respect to $XOR$ is $\frac{1}{2}$.*

*Proof.* For every $n \geq 2$, pair vector $Z$ and index $i \in [n]$, the output $O$ of XOR is independent of the $i$th input bit $D_i$, and hence also from the $i$th sampled bit $S_i$. Therefore, for every guesser $G$, every guess is independent of the sampled bit, and the success probability in every taken guess is $\frac{1}{2}$, that is, $Pr[S_i = T_i | T_i \neq 0] = \frac{1}{2}$, so the efficacy is $\frac{1}{2}$. $\square$

We deduce that ORA is not asymptotically tight for XOR.

**Corollary B.10** (ORA is Not Asymptotically Tight for XOR). *ORA is not asymptotically tight for XOR.*

*Proof.* Using Lemma A.4, since for every adversary strategy $\{f_n\}_{n \in \mathbb{N}}$, $E_{M,f_n,n} \xrightarrow{n \to \infty} \frac{1}{2} < p(\infty) = 1$. $\square$

The adversary's uncertainty about the other elements significantly reduces the efficacy of ORA with respect to XOR. Given an output of the algorithm, if the adversary has full knowledge of the other elements, it can determine the value of the specific element. On the other hand, if the adversary has only a uniform prior belief about the other elements, the element cannot be guessed better than at random.

Generally, the uncertainty of the adversary about the other elements may decrease the efficacy. The whole ORA process is an algorithm that first samples a database $D \sim \theta_Z$, and then runs the algorithm $M$ on this sampled dataset. The sampling adds another layer of privacy, so the privacy level of this algorithm may be better than that of $M$.

## C Efficacy and Asymptotic Tightness of ORA

Throughout this section we consider the probability distributions induced by the process $S \sim U^n, O \sim M(Z(S)), T = G(O)$, where $G$ is a maximum likelihood guesser (possibly with a threshold), and omit them from notation when clear from the context.

---

[16]Bhaskar et al. [17] show that XOR is noiseless private which implies this gap.

## C.1 Efficacy Without Abstentions

**Theorem C.1** (Optimal Efficacy Without Abstentions). *(Theorem 5.1) For every algorithm $M$ and a pair vector $Z$,*

$$E^*_{M,Z,n} = \varepsilon_{AE\text{-}AC}(M_Z, U^n) \overset{(1)}{\leq} \varepsilon_{AC}(M_Z, U^n) \overset{(2)}{\leq} p\left(\varepsilon_D(M_Z, U^n)\right) \overset{(3)}{\leq} p\left(\varepsilon(M_Z)\right),$$

*where $E^*_{M,Z,n}$ denotes the efficacy of the maximum likelihood guesser that takes all $n$ guesses.*

*Proof.* We consider auditing with the maximum likelihood guesser. From Bayes' law and the fact that $\Pr(S_i = 1) = \Pr(S_i = -1)$ we have for any $o \in \mathcal{O}, i \in [n]$,

$$\Pr\left(S_i = 1 \mid O = o\right) = \frac{\Pr\left(O = o \mid S_i = 1\right)}{\Pr\left(O = o \mid S_i = 1\right) + \Pr\left(O = o \mid S_i = -1\right)},$$

and

$$\Pr\left(S_i = -1 \mid O = o\right) = \frac{\Pr\left(O = o \mid S_i = -1\right)}{\Pr\left(O = o \mid S_i = 1\right) + \Pr\left(O = o \mid S_i = -1\right)}.$$

Using this identity we get

$$\begin{aligned}
\Pr\left(S_i = T_i \mid O = o\right) &= \max\{\Pr\left(S_i = 1 \mid O = o\right), \Pr\left(S_i = -1 \mid O = o\right)\} \text{ (guesser definition)} \\
&= \frac{\max\{\Pr\left(O = o \mid S_i = 1\right), \Pr\left(O = o \mid S_i = -1\right)\}}{\Pr\left(O = o \mid S_i = 1\right) + \Pr\left(O = o \mid S_i = -1\right)} \text{ (previous identity)} \\
&= p\left(|\ell_{M,Z,i}(O)|\right) \text{ (Lemma C.3)}
\end{aligned}$$

Using this identity we get

$$\begin{aligned}
E^*_{M,Z,n} &= \mathbb{E}\left[\frac{V_n}{R_n}\right] \\
&= \frac{1}{n}\sum_{i \in [n]} \Pr\left[S_i = T_i\right] \\
&= \frac{1}{n}\sum_{i \in [n]} \underset{O' \sim M_Z(U^n)}{\mathbb{E}} \left[\Pr\left[S_i = T_i \mid O = O'\right]\right] \text{ (Law of total expectation)} \\
&= \frac{1}{n}\sum_{i \in [n]} \underset{O' \sim M_Z(U^n)}{\mathbb{E}} \left[p\left(|\ell_{M,Z,i}(O')|\right)\right] \text{ (previous identity)} \\
&= \underset{O' \sim M_Z(U^n)}{\mathbb{E}} \left[\frac{1}{n}\sum_{i \in [n]} p\left(|\ell_{M,Z,i}(O')|\right)\right] \text{ (Linearity of expectation)} \\
&= \varepsilon_{AE\text{-}AC}(M_Z, U^n),
\end{aligned}$$

which completes the proof of the equality part.

The inequalities are because the privacy notions are ordered from the most relaxed to the strictest. All relaxations boil down to the fact that the average is bounded by the maximum. In the case of the first inequality the average is over $i$, in the second it is over $o$, and in the third it is over the sampling of the other elements. $\qquad\square$

### C.1.1 Connection to Total Variation

We present the max-divergence and the total variation distance. The max divergence measures the highest value of the log-likelihood ratio, and the total variation distance measures the total distance between the probability mass functions.

**Definition C.2** (Max Divergence, Total Variation Distance). Let $P$ and $Q$ be distributions over a discrete set $X$ represented by their probability mass functions.[17]

The max divergence of $P$ from $Q$ is

$$D_\infty(P||Q) := \max_{x \in X} \ln\left(\frac{P(x)}{Q(x)}\right).^{18}$$

The total variation distance between $P$ and $Q$ is

$$D_{\text{TV}}(P||Q) := \frac{1}{2} \sum_{x \in X} |P(X) - Q(X)|.$$

We prove two simple identities.

**Lemma C.3.** *Given distributions $P, Q$ over some domain $\mathcal{O}$ we have*

$$p\left(|\ell(o; P, Q)|\right) = \frac{\max\{P(o), Q(o)\}}{P(o) + Q(o)},$$

*and*

$$\mathop{\mathbb{E}}_{O \sim P}\left[p\left(|\ell(o; P, Q)|\right)\right] + \mathop{\mathbb{E}}_{O \sim Q}\left[p\left(|\ell(o; P, Q)|\right)\right] = 1 + \boldsymbol{D}_{TV}(P||Q),$$

*where $\ell(o; P, Q) := \log\left(\frac{P(o)}{Q(o)}\right)$ is the log probability ratio.*

*Proof.* We have

$$p\left(|\ell(o; P, Q)|\right) = \frac{e^{|\ell(o; P, Q)|}}{e^{|\ell(o; P, Q)|} + 1} = \begin{cases} \frac{P(o)}{P(o)+Q(o)} & P(o) > Q(o) \\ \frac{Q(o)}{P(o)+Q(o)} & P(o) \leq Q(o) \end{cases} = \frac{\max\{P(o), Q(o)\}}{P(o) + Q(o)}.$$

Combining this identity with the fact that $|x - y| + x + y = 2\max\{x, y\}$ we get

$$\begin{aligned}
\mathop{\mathbb{E}}_{O \sim P}\left[p\left(|\ell(o; P, Q)|\right)\right] + \mathop{\mathbb{E}}_{O \sim Q}\left[p\left(|\ell(o; P, Q)|\right)\right] &= \int_o (P(o) + Q(o)) \cdot p\left(|\ell(o; P, Q)|\right) do \\
&= \int_o (P(o) + Q(o)) \cdot \frac{\max\{P(o), Q(o)\}}{P(o) + Q(o)} do \\
&= \int_o \max\{P(o), Q(o)\} do \\
&= \frac{1}{2}\int_o P(o) + Q(o) + |P(o) - Q(o)| do \\
&= 1 + \boldsymbol{D}_{TV}(P||Q). \qquad \square
\end{aligned}$$

**Proposition C.4.** *For every algorithm $M$ and a pair vector $Z$,*

$$E^*_{M, Z, n} = \frac{1}{2} + \frac{1}{2} \cdot \frac{1}{n} \sum_{i \in [n]} \boldsymbol{D}_{TV}\left(M\left(U^n_{|D_i = -1}\right) \| M\left(U^n_{|D_i = 1}\right)\right),$$

*where $U^n_{|D_i = -1}$ is the distribution $U^n$ conditioned on the event $D_i = -1$ (and similarly for 1).*

---

[17]For simplicity, we present the definitions for the discrete case, but they can be extended to the continuous case by replacing probability mass functions with probability density functions, replacing sums with integrals, and handling zero-probability issues. We address this issue where it has implications.

[18]For the divergence definitions we define the log-likelihood ratio as above if both $P(x) \neq 0$ and $Q(x) \neq 0$. If only $Q(x) = 0$, then $L_{P||Q}(x) := \infty$, and if $P(x) = 0$, then $L_{P||Q}(x) := -\infty$.

*Proof.* We use the characterization of the optimal efficacy from the proof of Theorem C.1 and Lemma C.3 in the special case of the uniform prior to obtain

$$E^*_{M,Z,n} = \frac{1}{n} \sum_{i \in [n]} \mathop{\mathbb{E}}_{O \sim M_Z(U^n)} [p(|\ell_{M,Z,i}(O)|)] \text{ (from the proof of Theorem C.1)}$$

$$= \frac{1}{n} \sum_{i \in [n]} \left( \Pr[D_i = -1] \mathop{\mathbb{E}}_{O \sim M\left(U^n_{|D_i=-1}\right)} [p(|\ell_{M,Z,i}(O)|)] \right.$$

$$\left. + \Pr[D_i = 1] \mathop{\mathbb{E}}_{O \sim M\left(U^n_{|D_i=1}\right)} [p(|\ell_{M,Z,i}(O)|)] \right) \text{ (Law of total probability)}$$

$$= \frac{1}{2} \cdot \frac{1}{n} \sum_{i \in [n]} \left( \mathop{\mathbb{E}}_{O \sim M\left(U^n_{|D_i=-1}\right)} [p(|\ell_{M,Z,i}(O)|)] + \mathop{\mathbb{E}}_{O \sim M\left(U^n_{|D_i=1}\right)} [p(|\ell_{M,Z,i}(O)|)] \right) \text{ (the prior is uniform)}$$

$$= \frac{1}{2} \cdot \frac{1}{n} \sum_{i \in [n]} \left( 1 + \boldsymbol{D}_{TV} \left( M\left(U^n_{|D_i=-1}\right) \| M\left(U^n_{|D_i=1}\right) \right) \right) \text{ (Lemma C.3)}$$

$$= \frac{1}{2} + \frac{1}{2} \cdot \frac{1}{n} \sum_{i \in [n]} \boldsymbol{D}_{TV} \left( M\left(U^n_{|D_i=-1}\right) \| M\left(U^n_{|D_i=1}\right) \right). \qquad \square$$

## C.2 Efficacy With Abstentions

**Theorem C.5** (Optimal Efficacy). *(Theorem 5.2) For every algorithm $M$ and a pair vector $Z$,*

$$E^*_{M,Z,n,k} = \varepsilon^k_{AE\text{-}AC}(M_Z, U^n) \overset{(1)}{\le} \varepsilon_{AC}(M_Z, U^n) \overset{(2)}{\le} p(\varepsilon_D(M_Z, U^n)) \overset{(3)}{\le} p(\varepsilon(M_Z)).$$

*Proof.* The proof follows a similar structure to the case without abstentions, this time using the fact that $k$ guesses were made, and that $S_i = T_i$ implies $|T_i| = 1$.

$$E^*_{M,Z,n,k} = \mathbb{E} \left[ \frac{V_n}{R_n} \right]$$

$$= \mathop{\mathbb{E}}_{O' \sim M_Z(U^n)} \left[ \frac{1}{k} \sum_{i \in [n]} \Pr[S'_i = T'_i \mid O = O'] \right] \text{ (Guesser definition)}$$

$$= \mathop{\mathbb{E}}_{O' \sim M_Z(U^n)} \left[ \frac{1}{k} \sum_{i \in I_k(O')} \Pr[S'_i = T'_i \mid O = O'] \right] \text{ (Definition of $I_k$)}$$

$$= \mathop{\mathbb{E}}_{O' \sim M_Z(U^n)} \left[ \frac{1}{k} \sum_{i \in I_k(O')} p(|\ell_{M,Z,i}(O')|) \right] \text{ (Lemma C.3)}$$

$$= \varepsilon^k_{AE\text{-}AC}(M_Z, U^n)$$

The inequalities follow from the same argument as in the proof of Theorem C.1. $\qquad \square$

## C.3 Asymptotic Tightness Without Abstentions

We show that ORA without abstentions is asymptotically tight for an algorithm if and only if there exist pair vectors such that selecting the elements according to the pair vector and then applying the algorithm is "asymptotically post-process-able" to local randomized response.

**Lemma C.6** (Efficacy is mean of success probabilities). *For every randomized algorithm $M : X^n \to \mathcal{O}$, pair vector $Z = (x_1, y_1, ..., x_n, y_n) \in X^{2n}$ and guesser $G$,*

$$E_{M,Z,G,n} = \frac{1}{n} \sum_{i=1}^{n} \mathop{\mathbb{E}}_{S_{-i} \sim U^{n-1}} \left[ \mathop{\Pr}_{\substack{s_i \sim U^1 \\ T \sim A(S_{-i}, s_i)}} [T_i = s_i] \right],$$

*where $A_n = G_n \circ M_Z$.*

*Proof.*

$$E_{M,Z,G,n} = \mathbb{E}\left[\frac{V_n}{R_n}\right] \text{ (by definition)}$$

$$= \frac{1}{n}\mathbb{E}\left[V_n\right] \text{ (no abstentions, linearity of expectation)}$$

$$= \frac{1}{n}\sum_{i=1}^{n} \Pr_{\substack{S \sim U^1 \\ T \sim A_n(S)}}\left[T_i = S_i\right] \text{ (counting accurate guesses by indexes)}$$

$$= \frac{1}{n}\sum_{i=1}^{n} \Pr_{\substack{S \sim U^n \\ T \sim A_n(S)}}\left[T_i = s_i\right] \text{ (split to indexes, definition of sampling)}$$

$$= \frac{1}{n}\sum_{i=1}^{n} \mathop{\mathbb{E}}_{S_{-i} \sim U^{n-1}}\left[\Pr_{\substack{s_i \sim U^1 \\ T \sim A_n(S_{-i},s_i)}}\left[T_i = s_i\right]\right] \text{ (Law of total probability, using independence)}$$

$\square$

### C.3.1 The Case of Fixed n

We show that an algorithm with a fixed-size input can be audited by ORA without abstentions with perfect efficacy if and only if there exists a pair vector under which it can be post-processed to act like local randomized response.

**Lemma C.7.** *For every randomized algorithm $M : X^n \to \mathcal{O}$, pair vector $Z = (x_1, y_1, ..., x_n, y_n) \in X^{2n}$ and guesser $G$, $E_{M,Z,G,n} = p\left(\varepsilon(M)\right)$ if and only if $A = LRR_{\varepsilon(M)}$.*

*Proof.* Notice that $\varepsilon(A) \leq \varepsilon(M_Z) \leq \varepsilon(M)$, where the first inequality is from the post-processing property of differential privacy, and the second is because for every $i \in [n]$, the $i$th element in the output of $Z$ is determined by the $i$th element of its input.

For every index $i \in [n]$ and $S_{-i} \in \{-1, 1\}^{n-1}$, let $p_{i|S_{-i}} := \Pr_{\substack{s_i \sim U^1 \\ O \sim A(S_{-i},s_i)}}\left[O_i = s_i\right]$ denote the

success probability in guessing the $i$th index conditioned on the values of the other indices $S_{-i}$. Using Lemma C.6, $E_{M,Z,G,n} = \frac{1}{n}\sum_{i=1}^{n}\mathop{\mathbb{E}}_{S_{-i} \sim U^{n-1}}\left[p_{i|S_{-i}}\right]$. Using differential privacy, for every

index $i \in [n]$ and $S_{-i} \in \{-1, 1\}^{n-1}$, $p_{i|S_{-i}}$ is bounded by $p\left(\varepsilon(A)\right)$. Therefore, as a mean of bounded terms, $E_{M,Z,G,n} = p\left(\varepsilon(M)\right)$ if and only if $\varepsilon(A) = \varepsilon(M)$ and for every $i \in [n]$ and $S_{-i} \in \{-1, 1\}^{n-1}$, $p_{i|S_{-i}} = p\left(\varepsilon(M)\right)$.

Using differential privacy, for every $i \in [n]$ and $S_{-i} \in \{-1, 1\}^{n-1}$, $p_{i|S_{-i}} = p\left(\varepsilon(M)\right)$ if and only if for every $s_i \in \{-1, 1\}$, $A(S_{-i}, s_i) \stackrel{d}{=} RR_{\varepsilon(M)}(s_i)$, that is, for every $S \in \{-1, 1\}^n$, $A(S) \stackrel{d}{=} RR_{\varepsilon(M)}(S_i)$.

For every $i \in [n]$, let $X_i := \mathbb{1}_{A(S)_i = S_i}$ denote the random variable indicating whether the $i$th element in the output of $A$'s output matches the $i$th element of its input. For every $i \in [n]$, $X_i \sim \text{Ber}\left(p\left(\varepsilon(A)\right)\right)$. Using differential privacy, for every $i \in [n]$ and $x_{<i} \in \{0, 1\}^{i-1}$, $\Pr[X_i = 1 | X_{<i} = x_{<i}] \leq p\left(\varepsilon(M)\right)$. Hence, $X_1, ..., X_n \stackrel{\text{i.i.d.}}{\sim} \text{Ber}\left(p\left(\varepsilon(A)\right)\right)$, so $A = LRR_{\varepsilon(M)}$. $\square$

**Proposition C.8** (ORA has perfect efficacy iff Local Randomized Response Equivalent)**.** *For every $n \in \mathbb{N}$, randomized algorithm $M : X^n \to \mathcal{O}$, and pair vector $Z \in X^{2n}$, $E^*_{M,Z,n} = p\left(\varepsilon(M)\right)$ if and only if there exists some $f$ such that $f \circ M_Z = LRR_{\varepsilon(M)}$.*

*Proof.* We show that there exists a guesser $G : \mathcal{O} \to \{-1, 1\}$ such that $E_{M,Z,G,n} = p\left(\varepsilon(M)\right)$ if and only if there exists a randomized function $f : \mathcal{O} \to \{-1, 1\}$ such that, $f \circ M_Z \stackrel{d}{=} LRR_{\varepsilon(M)}$. By identifying guessers with such post-processing randomized functions, it is enough to show that for

every randomized function $G : \mathcal{O} \to \{-1, 1\}$, $E_{M,Z,G,n} = p\left(\varepsilon(M)\right)$ if and only if $A = LRR_{\varepsilon(M)}$. Lemma C.7 shows that and completes the proof. $\square$

### C.3.2 The Asymptotic Case

We extend proposition C.8 to the asymptotic case.

We say that an $\varepsilon$-differentially private randomized algorithm $A : \{-1,1\}^n \to \{-1,1\}^n$ is $(p, a)$- *probably approximately* $RR_\varepsilon$ in the $i$th index if under uniform distribution of the input, the probability of the $i$th entry of the output to equal the $i$th entry of the input is close to the maximal probability

achieved by $RR_\varepsilon$: $\Pr_{S_{-i} \sim U^{n-1}} \left[ \Pr_{\substack{s_i \sim U^1 \\ T \sim A(S_{-i}, s_i)}} [T_i = s_i] \geq p\left(\varepsilon\right) - a \right] \geq p$, and we denote this condition

by $A_i \overset{p,a}{\simeq} RR_\varepsilon$.

We say that a sequence of randomized algorithms $\{A_n : \{-1,1\}^n \to \{-1,1\}^n\}_{n \in \mathbb{N}}$ *approaches* $LRR_\varepsilon$ if the ratio of indexes for which it behaves like $RR_\varepsilon$ approaches 1; that is, if for every $p < 1$ and $a > 0$, $\frac{1}{n} \left| \left\{ i \in [n] : A_i \overset{p,a}{\simeq} RR_\varepsilon \right\} \right| \xrightarrow{n \to \infty} 1$, we denote this by $A_n \xrightarrow{n \to \infty} LRR_\varepsilon$.

**Lemma C.9.** *For every randomized algorithm* $M : X^n \to \{-1, 1\}^n$, *sequence of pair vectors* $\{Z_n \in X^{2n}\}_{n \in \mathbb{N}}$, *and sequence of randomized functions* $\{G_n : \mathcal{O} \to \{-1,1\}^n\}_{n \in \mathbb{N}}$, $E_{M,Z,f_n,n} \xrightarrow{n \to \infty} p\left(\varepsilon(M)\right)$ *if and only if* $f_n \circ M_{Z_n} \xrightarrow{n \to \infty} LRR_{\varepsilon(M)}$.

*Proof.* ($\Leftarrow$) We assume that $f_n \circ M_{Z_n} \xrightarrow{n \to \infty} LRR_{\varepsilon(M)}$. Let $0 < \delta < p\left(\varepsilon(M)\right)$. We define $m = p = \sqrt{\frac{p(\varepsilon(M)) - \delta}{p(\varepsilon(M)) - \frac{3\delta}{4}}}$ and $a := \frac{\delta}{2}$. There exists $N \in \mathbb{N}$ such that for every $n > N$, $f_n \circ M_{Z_n} \overset{m,p,a}{\simeq} LRR_{\varepsilon(M)}$, and hence

$$E_{M,Z_n,f_n,n} = \frac{1}{n} \sum_{i=1}^{n} \Pr_{\substack{S \sim U^1 \\ O \sim (f_n \circ M_{Z_n})(S)}} [O_i = S_i] \text{ (counting accurate guesses by indexes)}$$

$$= \frac{1}{n} \sum_{i=1}^{n} \mathbb{E}_{S_{-i} \sim U^{n-1}} \left[ \Pr_{\substack{s_i \sim U^1 \\ O \sim A(S_{-i}, s_i)}} [O_i = s_i] \right] \text{ (Law of total probability, using independence)}$$

$$\geq \frac{1}{n} mnp(p\left(\varepsilon(M)\right) - a) \text{ (Since } f_n \circ M_{Z_n} \overset{m,p,a}{\simeq} LRR_{\varepsilon(M)}\text{)}$$

$$= \left( \frac{p\left(\varepsilon(M)\right) - \delta}{p\left(\varepsilon(M)\right) - \frac{3\delta}{4}} \right) \left( p\left(\varepsilon(M)\right) - \frac{\delta}{2} \right) \text{ (substituting the values)}$$

$$= \frac{p\left(\varepsilon(M)\right) - \frac{\delta}{2}}{p\left(\varepsilon(M)\right) - \frac{3\delta}{4}} (p\left(\varepsilon(M)\right) - \delta) \text{ (algebra)}$$

$$\geq p\left(\varepsilon(M)\right) - \delta \text{ (the factor is lesser than 1).}$$

Therefore, $E_{M,Z_n,f_n,n} \xrightarrow{n \to \infty} p\left(\varepsilon(M)\right)$.

($\Rightarrow$) We show that if $\neg\ f_n \circ M_{Z_n} \overset{m,p,a}{\simeq} LRR_{\varepsilon(M)}$, then $\neg\ E_{M,Z_n,f_n,n} \xrightarrow{n \to \infty} p\left(\varepsilon(M)\right)$. If $\neg\ G^* \circ M_{Z_n} \overset{m,p,a}{\simeq} LRR_{\varepsilon(M)}$, that is, there exist $m < 1$, $p < 1$, and $a > 0$ and a sequence $\{n_k\}_{k \in \mathbb{N}}$ such that for every $k \in \mathbb{N}$,

$$\left| \left\{ i \in [n_k] : \Pr_{S_{-i} \sim U^{n-1}} \left[ \Pr_{\substack{s_i \sim U^1 \\ O \sim A(S_{-i}, s_i)}} [O_i = s_i] < p\left(\varepsilon\right) - a \right] > 1 - p \right\} \right| > (1 - m)n_k. \quad (1)$$

Therefore, for every $k \in \mathbb{N}$,

$$E_{M,Z_n,f_n,n} = \frac{1}{n_k} \sum_{i=1}^{n_k} \mathbb{E}_{S_{-i} \sim U^{n-1}} \left[ \Pr_{\substack{s_i \sim U^1 \\ O \sim A(S_{-i},s_i)}} [O_i = s_i] \right] \text{ (similarly to above)}$$

$$\leq mp\left(\varepsilon(M)\right) + (1-m)(p \cdot p\left(\varepsilon(M)\right) + (1-p)(p\left(\varepsilon(M)\right) - a)) \text{ (using Equation 1)}$$

$$= p\left(\varepsilon(M)\right) - (1-m)(1-p)a \text{ (algebra)}$$

$$< p\left(\varepsilon(M)\right),$$

and hence $\neg \ E_{M,Z_n,f_n,n} \xrightarrow{n\to\infty} p\left(\varepsilon(M)\right)$. $\qquad\square$

### C.3.3 Condition For Tightness Without Abstentions

**Theorem C.10** (ORA is asymptotically tight iff Approaches Local Randomized Response). *ORA without abstentions is asymptotically tight for a randomized algorithm $M : X^* \to \mathcal{O}$ with a sequence of pair vectors $\{Z_n \in X^{2n}\}_{n\in\mathbb{N}}$ if and only if there exists a sequence of randomized functions $\{f_n : \mathcal{O} \to \{-1,1\}^n\}_{n\in\mathbb{N}}$ such that*

$$f_n \circ M_{\mathcal{Z}_n} \xrightarrow{n\to\infty} LRR_{\varepsilon(M)},$$

*where $\mathcal{Z}_n$ is the function that maps a bit to its corresponding element in the pair vector $Z_n$.*

*Proof.* Using Lemma A.4, ORA without abstentions is asymptotically tight for $M$ with $\{Z_n \in X^{2n}\}_{n\in\mathbb{N}}$ if and only if there exists a sequence of guessers $\{G_n : \mathcal{O} \to \{-1,1\}\}_{n\in\mathbb{N}}$ such that $E_{M,Z_n,G_n,n} \xrightarrow{n\to\infty} p\left(\varepsilon(M)\right)$. We show it happens if and only if there exist a sequence of randomized functions $\{f_n : \mathcal{O} \to \{-1,1\}^n\}_{n\in\mathbb{N}}$ such that $f_n \circ M_{\mathcal{Z}_n} \xrightarrow{n\to\infty} LRR_{\varepsilon(M)}$. By identifying guessers with such post-processing randomized functions, it is enough to show that for every sequence of guessers $\{G_n : \mathcal{O} \to \{-1,1\}\}_{n\in\mathbb{N}}$, $E_{M,Z_n,G_n,n} \xrightarrow{n\to\infty} p\left(\varepsilon(M)\right)$ if and only if $G_n \circ M_{\mathcal{Z}_n} \xrightarrow{n\to\infty} LRR_{\varepsilon(M)}$. Lemma C.9 shows that and completes the proof. $\qquad\square$

### C.4 Asymptotic Tightness With Abstentions

**Theorem C.11** (Condition for Asymptotic Tightness of ORA). *(Proposition 5.3) ORA is asymptotically tight for a randomized algorithm $M : X^* \to \mathcal{O}$ and sequence of pair vectors $\{Z_n \in X^{2n}\}_{n\in\mathbb{N}}$ if and only if for every $\varepsilon' < \varepsilon(M)$,*

$$|\{i \in [n] : |\ell_{M,Z_n,i}| \geq \varepsilon'\}| \xrightarrow[n\to\infty]{P} \infty.$$

*Proof.* Using Lemma A.4, ORA is asymptotically tight for $M$ if and only if there exists an adversary strategy with unlimited guesses such that the efficacy approaches $p\left(\varepsilon(M)\right)$. It is enough to consider maximum likelihood guessers that commit to guess at least $k(n) \xrightarrow{n\to\infty} \infty$ guesses and check under which condition there exists such a guesser whose efficacy approaches $p\left(\varepsilon(M)\right)$. That is because any guesser with unlimited guesses can be converted to a guesser that commits to guess at least $k(n) \xrightarrow{n\to\infty} \infty$ guesses and its guesses distribution is arbitrarily close to the the original guesser's distribution.

$$E^*_{M,Z,n,k(n)} \xrightarrow{n\to\infty} p\left(\varepsilon(M)\right)$$

$$\iff \mathbb{E}_{O\sim M_Z(U^n)} \left[ \frac{1}{k(n)} \sum_{i \in I_{k(n)}(O)} p\left(|\ell_{M,Z_n,i}(O)|\right) \right] \xrightarrow{n\to\infty} p\left(\varepsilon(M)\right) \text{ (using Theorem 5.2)}$$

$$\iff \forall q < p\left(\varepsilon(M)\right) : \Pr_{O\sim M_Z(U^n)} \left[ \frac{1}{k(n)} \sum_{i \in I_{k(n)}(O)} p\left(|\ell_{M,Z_n,i}(O)|\right) \geq q \right] \xrightarrow{n\to\infty} 1 \text{ ($p\left(|\ell_{M,Z_n,i}(O)|\right)$ is bounded)}$$

$$\iff \forall q < p\left(\varepsilon(M)\right) : |\{i \in [n] : p\left(|\ell_{M,Z_n,i}(O)|\right) \geq q\}| \xrightarrow[n\to\infty]{P} \infty \text{ (using $k(n) \xrightarrow{n\to\infty} \infty$ )}$$

$$\iff \forall \varepsilon' < \varepsilon(M) : |\{i \in [n] : |\ell_{M,Z_n,i}(O)| \geq \varepsilon'\}| \xrightarrow[n\to\infty]{P} \infty \text{ (using the monotonicity of $p$).}$$

$\qquad\square$

## C.5 Local Algorithms

The Laplace algorithm [13] is a popular privacy-preserving algorithm. We calculate the optimal efficacy of ORA without abstentions for the algorithm that applies the Laplace algorithm independently to each element of a dataset, and show it displays a significant auditing gap for ORA due to the non-worst-case outputs gap.

**Definition C.12** (Local Laplace [13])**.** The Local Laplace algorithm $LLP_\varepsilon : \{-1, 1\}^n \to \mathbb{R}^n$ is an algorithm parametrized by $\varepsilon > 0$ that takes elements in $\{-1, 1\}$ as input and adds Laplace-distributed noise to each one.

$$LLP(D) = (LP(D_1), ..., LP(D_n)),$$

where

$$LP(x) = x + Laplace\left(b = \frac{2}{\varepsilon}\right).$$

We calculate the optimal efficacy of ORA without abstentions for the Local Laplace algorithm.

**Proposition C.13.** *For every $\varepsilon > 0$, the optimal efficacy of ORA without abstentions for $LLP_\varepsilon$ is $1 - \frac{1}{2}exp\left(-\frac{\varepsilon}{2}\right)$.*

*Proof.* The size of the domain of the algorithm is 2, and hence all of the pair vectors are equivalent. We consider the pair vector $Z = (-1, 1, ..., -1, 1) \in \{-1, 1\}^{2n}$. The optimal efficacy without abstentions for $Z$ is achieved by the maximum likelihood guesser $G^*_{LLP_\varepsilon, Z}$. It guesses as follows.

$$G^*_{LLP_\varepsilon, Z}(O)_i := \begin{cases} -1 & \text{if } \Pr[LP(-1) = O] \geq \Pr[LP(1) = O] \\ 1 & \text{else} \end{cases} \text{(Maximum likelihood guesser)}$$

$$= \begin{cases} -1 & \text{if } O \leq 0 \\ 1 & \text{else} \end{cases} \text{(using Laplace distribution's PMF).}$$

For every $i \in [n]$, the probability of $G^*_{LLP_\varepsilon, Z}$ to accurately guess the $i$th element is

$$\begin{aligned}
\Pr[T_i = S_i | T_i \neq 0] &= \Pr[T_i = S_i] \text{ (no abstentions)} \\
&= \Pr[S_i = -1]\Pr[T_i = S_i | S_i = -1] + \Pr[S_i = 1]\Pr[T_i = S_i | S_i = 1] \text{ (Law of total probability)} \\
&= \frac{1}{2}(\Pr[T_i = S_i | S_i = -1] + \Pr[T_i = S_i | S_i = 1]) \text{ (uniform sampling)} \\
&= \frac{1}{2}(\Pr[L(-1) < 0] + \Pr[L(1) > 0]) \text{ (by the behavior of the maximum likelihood guesser)} \\
&= \Pr[L(-1) < 0] \text{ (using the symmetry of Laplace distribution)} \\
&= \Pr\left[-1 + Lap\left(b = \frac{2}{\varepsilon}\right) < 0\right] \text{ (using the Laplace algorithm definition)} \\
&= 1 - \frac{1}{2}exp\left(-\frac{\varepsilon}{2}\right) \text{ (using the Laplace distribution's CDF formula).}
\end{aligned}$$

Using the linearity of expectation, the efficacy is $1 - \frac{1}{2}exp\left(-\frac{\varepsilon}{2}\right)$. $\qquad\square$

Figure 4 shows $p^{-1}\left(E^*_{LLP_\varepsilon}\right)$ for multiple values of $\varepsilon$. Since this is a local algorithm, the bounds of the privacy level converge to this quantity (see Lemma C.14).

**Lemma C.14** (Bounds Approach Privacy Level Corresponding to Efficacy for Local Algorithms)**.** *For every local randomized algorithm $M : X^n \to \mathcal{O}$, $x, y \in X$, and $\beta \in [0, 1)$, the resulting bounds from ORA of $M$ with $Z = (x, y, \ldots, x, y) \in X^{2n}$ and a maximum likelihood guesser without abstentions converge in probability to the privacy level corresponding to the efficacy,*

$$\left|\varepsilon'_\beta(V_n, R_n) - p^{-1}\left(E^*_{M, Z_n, n}\right)\right| \xrightarrow[n\to\infty]{P} 0.$$

*Proof.* By the locality of $M$, the events of accurately guessing different elements have equal probabilities and are independent. Hence, there exists $q \in [0, 1]$ such that $V_n \sim Bin(n, q)$. The guesser does not abstain so the accuracy is distributed as $\frac{V_n}{R_n} \sim \frac{1}{n}Bin(n, q)$.

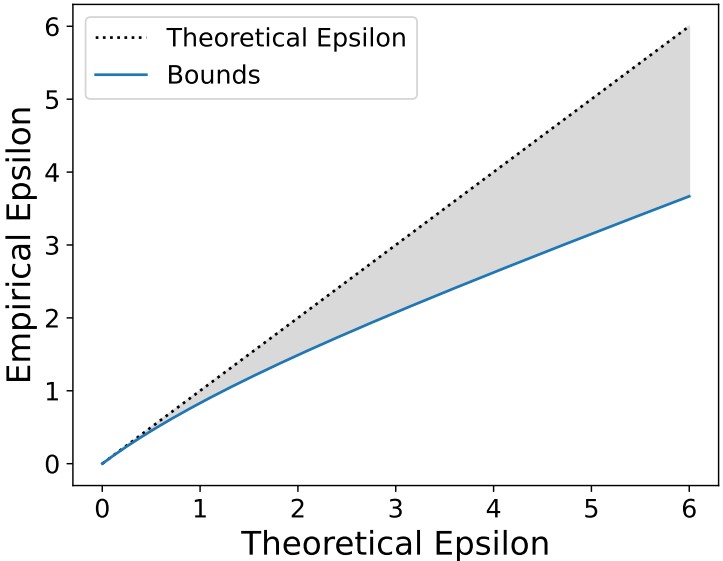

Figure 4: Bounds of ORA without abstentions of Local Laplace for multiple values of $\varepsilon$

Therefore,

$$
\begin{aligned}
\left| p\left(\varepsilon'_\beta(V_n, R_n)\right) - E^*_{M,Z_n,n} \right| &= \left| CPL\left(R_n, V_n, \beta\right) - \mathbb{E}\left[\frac{V_n}{R_n}\right] \right| \\
&\leq \left| CPL\left(R_n, V_n, \beta\right) - \frac{V_n}{R_n} \right| + \left| \mathbb{E}\left[\frac{V_n}{R_n}\right] - \frac{V_n}{R_n} \right| \\
&\leq \left| CPL\left(R_n, V_n, \beta\right) - \frac{V_n}{R_n} \right| + \left| q - \frac{V_n}{n} \right| \\
&\xrightarrow[n\to\infty]{P} 0 + 0 \text{ (using Lemma A.11 and the weak law of large numbers)} \\
&= 0.
\end{aligned}
$$

Using the continuity of $p^{-1}$,

$$
\left| \varepsilon'_\beta(V_n, R_n) - p^{-1}\left(E^*_{M,Z_n,n}\right) \right| \xrightarrow[n\to\infty]{P} 0. \qquad \square
$$

# D  DP-SGD: Case Study of ORA

## D.1  DP-SGD

Algorithm 3 is pseudocode of the DP-SGD algorithm.

## D.2  Theoretical Analysis

### D.2.1  Count and Symmetric Algorithms on $\{0, 1\}^n$

We bound the efficacy of ORA for symmetric binary algorithms on $\{0, 1\}^n$. This is a fundamental family of algorithms that includes any counting algorithm.

First, we consider the count algorithm that outputs the number of ones in the dataset. Counting queries serve as a basic building block in many algorithms. Un-noised counting queries are not differentially private for any finite $\varepsilon$. We show that ORA is not asymptotically tight for such queries. Moreover, we show that the optimal efficacy of ORA with respect to such queries, even with abstentions, approaches the minimal $\frac{1}{2}$ efficacy.

**Algorithm 3** DP-SGD - Differentially Private Stochastic Gradient Descent
___

1: **Input:** Training data $X \in \mathcal{X}^n$, loss function $l : \mathbb{R}^d \times \mathcal{X} \to \mathbb{R}$
2: **Parameters:** Number of steps $T \in \mathbb{N}$, sample rate $r \in (0, 1]$, clipping threshold $c > 0$, noise scale $\sigma > 0$, learning rate $\eta > 0$
3: Initialize model weights $w_0 \in \mathbb{R}^d$.
4: **for** $t = 1, \ldots, T$ : **do**
5:      Sample a batch $B$ from $X$ with sampling probability $r$.
6:      Compute the batch gradients $g_1, ..., g_{|B|} \in \mathbb{R}^d$ of $l$ with respect to $w_{t-1}$.
7:      Clip each gradient $\hat{g}_i = \min \left\{ 1, \frac{c}{\|g_i\|_2} \right\} \cdot g_i$.
8:      Compute a noisy multi-dimensional sum of the clipped gradients $\tilde{g} = \sum_{i=1}^{b} \hat{g}_i + \mathcal{N}(0, \sigma^2 c^2 I)$.
9:      Update the model weights $w_t = w_{t-1} - \eta \cdot \tilde{g}$.
10: **end for**
11: **Return:** All intermediate model weights $w_0, \ldots, w_T$.
___

Consider the count algorithm $C^n : \{0, 1\}^n \to \{0, ..., n\}$ which outputs the number of ones in the dataset, $C(D_1, ..., D_n) = |\{i : D_i = 1\}|$. The efficacy gap of ORA for the count algorithm is due to a combination of the non-worst-case outputs and the interference gaps; namely, because with high probability, the output of the counting query does not reveal much about each element without knowing the true values of the other elements. With high probability, the output of the algorithm is close to $\frac{n}{2}$, and in this case, the probability of an optimal guesser to accurately guess any particular element is close to $\frac{1}{2}$. We show that the expected efficacy is the normalized mean absolute deviation from the expectation of the binomial distribution.

We use the following lemma about the mean absolute deviation of the binomial distribution to show that the optimal efficacy approaches $\frac{1}{2}$ when $n \to \infty$.

**Lemma D.1** (Bound on the mean absolute deviation of a binomial [19]). *For every $n \in \mathbb{N}$ and $p \in [0, 1]$,*

$$\mathbb{E}_{O \sim Bin(n,p)} [|O - np|] \leq \sqrt{np(1 - p)}.$$

**Proposition D.2** (ORA optimal efficacy for count approaches $\frac{1}{2}$). *For every sequence of pair vectors $Z = \{Z_n \in X^{2n}\}_{n \in \mathbb{N}}$, and number of guesses $k \in \mathbb{N}$, the optimal efficacy of ORA with abstentions for the count algorithm $C^n$ approaches $\frac{1}{2}$; that is, $E^*_{C^n, Z, n, k} \xrightarrow{n \to \infty} \frac{1}{2}$.*

*Proof.* Since $|X| = 2$, all pair vectors induce the same dataset distribution, so the optimal efficacy does not depend on the pair vector. Hence, it suffices to prove the claim for $Z = (0, 1, ..., 0, 1) \in X^{2n}$. For every $i \in [n]$ and $o \in \{0, ..., n\}$, the number of ones in the dataset except $D_i$ is distributed as $C \sim Bin\left(n - 1, \frac{1}{2}\right)$, and hence

$$p\left(|\ell_{M,Z,i}(o)|\right) = \frac{\max(\{Pr[O = o|S_i = -1], Pr[O = o|S_i = 1]\})}{Pr[O = o|S_i = -1] + Pr[O = o|S_i = 1]} \quad \text{(using Lemma C.3)}$$
$$= \frac{\max(\{Pr[C = o], Pr[C = o - 1]\})}{Pr[C = o] + Pr[C = o - 1]}.$$

For every $o \in \{1, ..., n\}$,

$$= \frac{\max(\{\binom{n-1}{o}(\frac{1}{2})^{n-1}, \binom{n-1}{o-1}(\frac{1}{2})^{n-1}\})}{\binom{n-1}{o}(\frac{1}{2})^{n-1} + \binom{n-1}{o-1}(\frac{1}{2})^{n-1}} \text{ (using the binomial distribution's PMF)}$$

$$= \frac{\max(\{\binom{n-1}{o}, \binom{n-1}{o-1}\})}{\binom{n-1}{o} + \binom{n-1}{o-1}}$$

$$= \frac{\max(\{\frac{(n-1)!}{o!(n-o-1)!}, \frac{(n-1)!}{(o-1)!(n-o)!}\})}{\frac{(n-1)!}{o!(n-o-1)!} + \frac{(n-1)!}{(o-1)!(n-o)!}}$$

$$= \frac{1}{n} max(\{o, n-o\}) \text{ (by algebraic manipulation)}$$

$$= \frac{1}{n}\left(\frac{n}{2} + \left|o - \frac{n}{2}\right|\right)$$

$$= \frac{1}{2} + \frac{1}{n}\left|o - \frac{n}{2}\right|.$$

Also for $o = 0$, $p(|\ell_{M,Z,i}(o)|) = 1 = \frac{1}{2} + \frac{1}{n}|o - \frac{n}{2}|$. Thus, for every $o \in \{0, ..., n\}$, $p(|\ell_{M,Z,i}(o)|) = \frac{1}{2} + \frac{1}{n}|o - \frac{n}{2}|$.

Therefore, for any number of guesses $k$, the optimal efficacy is

$$E^*_{C^n,Z,n,k} = \mathop{\mathbb{E}}_{O \sim \text{Bin}(n,\frac{1}{2})}\left[\frac{1}{k}\sum_{i \in I_k(O)} p(|\ell_{M,Z,i}(O)|)\right] \text{ (using Theorem 5.2)}$$

$$= \mathop{\mathbb{E}}_{O \sim \text{Bin}(n,\frac{1}{2})}\left[\frac{1}{2} + \frac{1}{n}\left|O - \frac{n}{2}\right|\right] \text{ (using the calculation above)}$$

$$= \frac{1}{2} + \frac{1}{n} \cdot \mathop{\mathbb{E}}_{O \sim \text{Bin}(n,\frac{1}{2})}\left[\left|O - \frac{n}{2}\right|\right]$$

$$\leq \frac{1}{2} + \frac{1}{n}\sqrt{\frac{n}{4}} \text{ (using Lemma D.1)}$$

$$= \frac{1}{2} + \frac{1}{2\sqrt{n}}$$

$$\xrightarrow{n \to \infty} \frac{1}{2}.$$

For every number of elements and number of guesses, the optimal efficacy $E^*_{C^n,Z,n,k}$ is at least $\frac{1}{2}$, so using the bound from above, the optimal efficacy with $k$ guesses approaches $\frac{1}{2}$, that is, $E^*_{C^n,Z,n,k} \xrightarrow{n \to \infty} \frac{1}{2}$. $\square$

We deduce that the optimal efficacy of ORA with respect to any symmetric algorithm approaches $\frac{1}{2}$. Using Lemma A.4, it follows that ORA is not asymptotically tight for any such algorithm with non-trivial privacy guarantees.

**Corollary D.3.** *For every symmetric algorithm $M : \{0,1\}^* \to \mathcal{O}$, sequence of pair vectors $Z = \{Z_n \in X^{2n}\}_{n \in \mathbb{N}}$, and number of guesses $k \in \mathbb{N}$, the optimal efficacy of ORA with abstentions for $M$ approaches $\frac{1}{2}$, that is, $E^*_{M,Z,n,k} \xrightarrow{n \to \infty} \frac{1}{2}$.*

*Proof.* Every symmetric algorithm $M : \{0,1\}^* \to \mathcal{O}$ is a post-processing of the count algorithm, and hence the efficacy of ORA for it is less than or equal to its efficacy for count. $\square$

### D.2.2 Count in Sets

We analyze auditing of the Count-In-Sets algorithm (see Definition 6.1), a simplistic algorithm that enable us to capture the essence of auditing DP-SGD, and show a condition for the tightness of ORA for it.

First, we prove a lemma about the distribution of the privacy loss of the Count algorithm which we use in the proof of the condition for asymptotic tightness of ORA of Count-In-Sets.

**Lemma D.4** (Privacy Loss of Count). *For every sequence of pair vectors $Z = \{Z_n \in X^{2n}\}_{n\in\mathbb{N}}$, index $i \in [n]$ and threshold $a > 0$, in ORA of Count with $Z$, the probability of the distributional privacy loss at the $i$th index to be at least $a$ can be characterized using the probability of deviation of the output $O$ from its mean as*

$$Pr[|\ell_{C^n,Z,i}| \geq a] = Pr\left[\left|O - \frac{n}{2}\right| \geq \frac{e^a - 1}{2(e^a + 1)}n\right].$$

*Proof.* Since $|X| = 2$, all pair vectors induce the same dataset distribution, so the distribution of the distributional privacy loss does not depend on the pair vector. Therefore it is enough to prove the claim for $Z = (0, 1, ..., 0, 1) \in X^{2n}$. For every $i \in [n]$ and $o \in \{0, ..., n\}$, the number of ones in the dataset except $D_i$ is distributed as $C \sim \text{Bin}\left(n - 1, \frac{1}{2}\right)$. We evaluate the absolute value of the distributional privacy loss.

For every $o \neq 0$,

$$|\ell_{C^n,Z,i}(o)| = \left|\ln\left(\frac{Pr[O = o|D_i = 0]}{Pr[O = o|D_i = 1]}\right)\right|$$
$$= \left|\ln\left(\frac{Pr[C = o]}{Pr[C = o - 1]}\right)\right|$$
$$= \left|\ln\left(\frac{\binom{n-1}{o}(\frac{1}{2})^{n-1}}{\binom{n-1}{o-1}(\frac{1}{2})^{n-1}}\right)\right|$$
$$= \left|\ln\left(\frac{(n-1)!(o-1)!(n-o)!}{o!(n-o-1)!(n-1)!}\right)\right|$$
$$= \left|\ln\left(\frac{n-o}{o}\right)\right|,$$

and for $o = 0$, $|\ell_{C^n,Z,i}(o)| = \left|\ln\left(\frac{Pr[C=o]}{Pr[C=o-1]}\right)\right| = \infty$, which is consistent with the result above by defining $\frac{x}{0} := \infty$.

For every $a > 0$, since $O \sim \text{Bin}\left(n, \frac{1}{2}\right)$,

$$Pr[|\ell_{C^n,Z,i}(O)| \geq a] = Pr\left[\left|\ln\left(\frac{n-O}{O}\right)\right| \geq a\right] \quad \text{(the calculation above)}$$
$$= Pr\left[\frac{n-O}{O} \leq e^{-a} \ \vee \ \frac{n-O}{O} \geq e^a\right] \quad \text{(ln's monotonicity)}$$
$$= Pr\left[O \leq \frac{n}{e^a + 1} \ \vee \ O \geq \frac{n}{e^{-a} + 1}\right] \quad \text{(algebra)}$$
$$= Pr\left[\left|O - \frac{n}{2}\right| \geq \frac{e^a - 1}{2(e^a + 1)}n\right] \quad \text{(algebra)}. \qquad \square$$

We characterize the condition for asymptotic tightness of ORA for Count-In-Sets; namely, we find a threshold on the size of the sets $s(n)$ that determines whether ORA is tight.

**Proposition D.5** (Condition for Asymptotic Tightness of ORA for Count-in-Sets). *ORA is asymptotically tight for $CIS_s^n$ if and only if $s(n) = o(log(n))$.*

*Proof.* Since $|X| = 2$, all pair vectors induce the same dataset distribution, so the optimal efficacy does not depend on the pair vector. Therefore, it is enough to prove the claim for $Z = \{Z_n = (0, 1, ..., 0, 1) \in X^{2n}\}_{n\in\mathbb{N}}$. Throughout the proof, we assume for convenience without loss of generality that $n$ is a multiple of $s(n)$.

For every $n > 0$ and $s \in [n]$, $CIS_s^n$ is a deterministic algorithm and $\varepsilon(CIS_s^n) = \infty$. Using Theorem 5.3, ORA is asymptotically tight for $CIS_s^n$ with $Z$ if and only if for every $a < \infty$,

$$|\{i \in [n] : |\ell_{M,Z,i}| \geq a\}| \xrightarrow[n\to\infty]{P} \infty.$$

We fix a threshold $a > 0$ on the privacy loss. In each set there are $s(n)$ elements and for every output, their privacy losses are equal. We denote by $A_j$ the event in which the elements in the $j$th set have privacy loss of at least $a$. Notice that for every $j, j' \in \left[\frac{n}{s(n)}\right]$, $A_j$ and $A_{j'}$ are independent and have equal probabilities, we denote this probability by $q(n)$.

The number of sets whose elements have privacy loss of at least $a$ is distributed as $\text{Bin}\left(\frac{n}{s(n)}, q(n)\right)$, and hence the number of elements with privacy loss of at least $a$ is distributed as $s(n) \cdot \text{Bin}\left(\frac{n}{s(n)}, q(n)\right)$. Using the weak law of large numbers,

$$|\{i \in [n] : |\ell_{M,Z,i}| \geq a\}| \xrightarrow[n\to\infty]{P} \infty \iff s(n) \cdot \text{Bin}\left(\frac{n}{s(n)}, q(n)\right) \xrightarrow[n\to\infty]{P} \infty$$

$$\iff s(n) \cdot \frac{n}{s(n)} \cdot q(n) \xrightarrow{n\to\infty} \infty$$

$$\iff n \cdot q(n) \xrightarrow{n\to\infty} \infty.$$

Using the previous lemma, since the Count-In-Sets algorithm is composed of multiple independent instances of the Count algorithm, each with input of $s(n)$ elements, for every $j \in \left[\frac{n}{s(n)}\right]$,

$$q(n) = Pr\left[\left|\ln\left(\frac{s(n) - O_j}{O_j}\right)\right| \geq a\right] = Pr\left[\left|O_j - \frac{s(n)}{2}\right| \geq \frac{e^a - 1}{2(e^a + 1)}s(n)\right],$$

where the counts of the sets are distributed as $O_1, ..., O_{\frac{n}{s(n)}} \overset{\text{i.i.d.}}{\sim} \text{Bin}\left(s(n), \frac{1}{2}\right)$.

We use Hoeffding's inequality to bound $q(n)$ from above

$$q(n) \leq 2exp\left(-\frac{e^{2a} - 2e^a + 1}{2(e^{2a} + 2e^a + 1)}s(n)\right).$$

We bound the term from below

$$q(n) \geq Pr\left[\left|O_j - \frac{s(n)}{2}\right| \geq \frac{1}{2}s(n)\right]$$

$$= Pr[O_j \in \{0, s(n)\}]$$

$$= 2^{-(s(n)-1)}.$$

Hence, for every $a > 0$,

$$2^{-(s(n)-1)} \leq q(n) \leq 2exp\left(-\frac{e^{2a} - 2e^a + 1}{2(e^{2a} + 2e^a + 1)}s(n)\right)$$

$$\Rightarrow -\log(q(n)) = \Theta(s(n)).$$

We find the condition for the convergence

$$nq(n) \xrightarrow{n\to\infty} \infty \iff q(n) = \omega\left(\frac{1}{n}\right)$$

$$\iff 2^{-s(n)} = \omega\left(\frac{1}{n}\right)$$

$$\iff s(n) = o\left(-\log\left(\frac{1}{n}\right)\right)$$

$$\iff s(n) = o(\log(n)).$$

Hence, ORA is asymptotically tight for $CIS_s^n$ if and only if $s(n) = o(\log(n))$. □

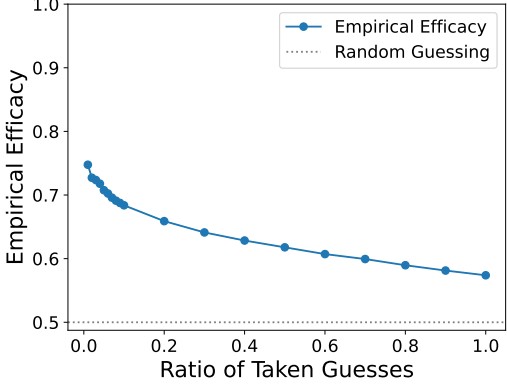 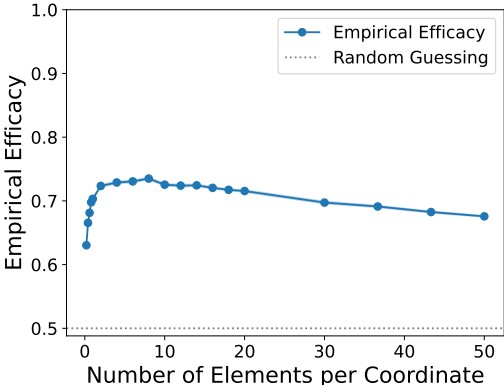

Figure 5: Effect of the fraction of taken guesses $\frac{k}{n}$ on ORA's empirical efficacy for $n = 5000$ elements.

Figure 6: Effect of the number of elements per coordinate $\frac{n}{d}$ on ORA's empirical efficacy when making $k = 100$ guesses.

## D.3 Experiments

Figures 5 and 6 show the empirical efficacy in the same experiments as Figures 1 and 2 respectively. Figure 5 shows that the efficacy decreases as the ratio of taken guesses increases. Figure 6 shows the efficacy tradeoff as the number of elements per coordinate varies. As expected, the trend in each figure is similar to the trend of the privacy estimations in the corresponding figure.

Figures 7 and 8 show results of experiments for multiple values of $\varepsilon$ to allow better comparison with the results of Steinke et al. [1]. Figure 7 shows the effect of the number of taken guesses and Figure 8 shows the effect of the number of elements. In these figures, each point represents the mean of the bounds in 20 experiments.

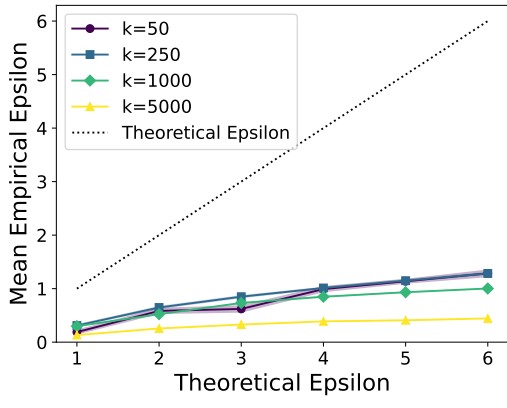 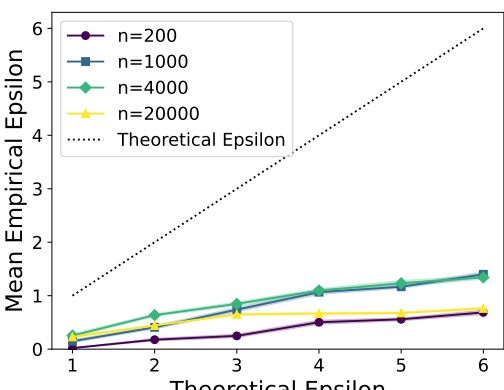

Figure 7: Effect of the number of taken guesses $k$ on ORA's results for multiple values of $\varepsilon$ for $n = 5000$.

Figure 8: Effect of the number of elements $n$ on ORA's results for multiple values of $\varepsilon$ for $k = 100$.

## E Adaptive ORA

In this section, we introduce Adaptive ORA (AORA), a valid and more effective variant of ORA in which the guesser $G$ guesses adaptively by using the true value of the sampled bits from $S$ it has already guessed. The AORA guesser has more knowledge about the other elements so the interference gap is reduced and the efficacy increases.

In Section 4 we discuss the interference gap, that is, we see that the efficacy of ORA is limited because the guesser lacks information about the other elements when it guesses one of them. We demonstrate this gap via the XOR algorithm ,for which ORA has minimal efficacy even though the algorithm is not differentially private. In Section 5, we formalize this idea by reasoning about the noiseless privacy loss, which models the adversary's uncertainty using a distribution over datasets. We show two variants of ORA that aim to improve this gap; the first, full-knowledge ORA, completely resolves this gap but it is not valid and hence not useful, and the second, Adaptive ORA, reduces the gap and is valid. We demonstrate the efficacy-gain using the crafted Xor-in-Pairs algorithm, for which ORA has minimal efficacy, while Adaptive ORA has perfect efficacy and unlimited guesses. In addition, we show the advantage of AORA for a more realistic algorithm, discussing AORA of Count-In-Sets, which provides a simple model of auditing DP-SGD.

## E.1 Full-Knowledge ORA (an Invalid Auditing Method)

For exposition, consider "Full-Knowledge ORA," a variant of ORA in which the guesser is exposed to all the other elements when it guesses one of them. In this variant, in each round $i \in [n]$, the guesser guesses the $i$th element based not only on the output $O$, but also on the values of the other elements $D^{-i} := D_1, ..., D_{i-1}, D_{i+1}, ..., D_n$. This auditing method perfectly matches the knowledge of the adversary in the differential privacy threat model, and so the interference gap is entirely resolved.

However, full-knowledge ORA is not valid (see Definition A.1 and Lemma A.7). Even though for every $\varepsilon$-differentially private algorithm, the success probability of each guess is bounded by $p(\varepsilon)$, the dependence between the successes is not limited and hence the number of accurate guesses $V$ is not stochastically dominated by $Binomial(r, p(\varepsilon))$, where $r$ is the number of taken guesses. Therefore, the validity proof of ORA which is based on this stochastic domination (see Lemma A.7) does not hold here.

We show an algorithm that demonstrates the invalidity of "Full-Knowledge ORA."

**Definition E.1** (XOR + Randomized Response (XRR)). The "XOR + Randomized Response" algorithm $XRR_\varepsilon^n : \{0,1\}^n \to \{0,1\}$ is the algorithm that computes the XOR of the bits it takes as input, and outputs an $\varepsilon$-randomized response of the XOR.

$$XRR_\varepsilon^n(D) = \begin{cases} D_1 \oplus ... \oplus D_n & \text{with probability } p(\varepsilon) \\ 1 - (D_1 \oplus ... \oplus D_n) & \text{with probability } 1 - p(\varepsilon) \end{cases}.$$

$XRR_\varepsilon^n$ is $\varepsilon$-differentially private. We show a guesser for which dependence between the successes of the different elements is maximal.

**Proposition E.2.** *In full-knowledge ORA, the number of accurate guesses $V$ with the pair vector $Z = (0, 1, ..., 0, 1) \in \{0,1\}^{2n}$ and the guesser*

$$G(i, O, S^{-i}) = XOR(S^{-i}) \oplus O$$

*is distributed as follows:*

$$V = \begin{cases} n & \text{with probability } p(\varepsilon) \\ 0 & \text{otherwise} \end{cases} \succcurlyeq Bin(n, p(\varepsilon)).$$

*Proof.* This follows from the fact that for every index $i \in [n]$, $G$ succeeds in guessing the $i$th element if and only if the output was the XOR. $\qquad\square$

We conclude that full-knowledge ORA is not valid. We might nonetheless hope to find another method that reduces the interference gap, while limiting the dependence between the successes in guessing the different elements.

## E.2 Adaptive ORA is Valid

We introduce Adaptive ORA, an auditing method that reduces the interference gap and increases the efficacy of one-run auditing, and prove its validity.

**Definition E.3** (Adaptive ORA (AORA)). Adaptive ORA works similarly to ORA, where the only difference is in the guessing phase. In ORA, for every $k \in [n]$, $G$ selects an index $I_k \in [n]$ that it still has not guessed, takes the output $O$ and the true values of the elements it already guessed $S_{I_1}, ..., S_{I_{k-1}}$ as input, and outputs $T_{I_k}$.

We present a claim presented by Steinke et al. [1]. This claim is used in the induction step of the validity proof of ORA.

**Lemma E.4** (Stochastic Dominance of Sum [1]). *For every set of random variables $X_1$, $X_2$, $Y_1$, and $Y_2$, if $X_1$ is stochastically dominated by $Y_1$, and for every $x \in \mathbb{R}$, $X_2|X_1 = x$ is stochastically dominated by $Y_2$, then $X_1 + X_2$ is stochastically dominated by $Y_1 + Y_2$.*

We show that the number of accurate guesses in Adaptive ORA is stochastically dominated by $\text{Bin}(n, p(\varepsilon))$, and hence it is valid. The proof is very similar to the analogous proof for ORA by [1] because the original proof bounds the success probability of every guess conditioned on the previously sampled bits.

**Proposition E.5** (Adaptive ORA is valid). *Adaptive ORA is a valid guessing-based auditing method.*

*Proof.* We follow the proof of the validity of ORA by Steinke et al. [1] and modify it slightly to allow the guesser to choose the guessing order.

Fix some $n \in \mathbb{N}$. We prove by induction on the number $k$ of guesses made that the number of accurate guesses until the $k$'th guess conditioned on the guess, the indexes chosen, and the sampled bits revealed so far is stochastically dominated by a binomial distribution, $V_k|T = t \preccurlyeq \text{Bin}\left(\|t_{i_{\leq k}}\|_1, p(\varepsilon(M))\right)$, where for any vector $a$, we denote by $a_{i_{<k}}$ and $a_{i_{\leq k}}$, $a_{i_1}, ..., a_{i_{k-1}}$ and $a_{i_1}, ..., a_{i_k}$, respectively. The claim trivially holds for $k = 0$ because no guesses have been made up to this point. We prove the induction step; that is, if the claim holds for a $k \in \{0, ..., n-1\}$, it also holds for $k + 1$.

We fix a guessing step $k \in [n]$ and calculate the sampled bit $S_{i_k}$ distribution given a guess $t \in \{-1, 1\}^n$, an ordering of guessing $i = (i_1, ..., i_n) \subseteq [n]$ chosen, and the sampled bits revealed so far $s_{i_{<k}} \in \{-1, 1\}^{k-1}$. The probability of the sampled bit to be $-1$ is

$$Pr[S_{i_k} = -1|T = t, I = i, S_{i_{<k}} = s_{i_{<k}}]$$

$$= \frac{Pr[T = t, I = i|S_{i_k} = -1, S_{i_{<k}} = s_{i_{<k}}]Pr[S_{i_k} = 1|S_{i_{<k}} = s_{i_{<k}}]}{Pr[T = t, I = i|S_{i_{<k}} = s_{i_{<k}}]} \text{ (Bayes' law)}$$

$$= \frac{Pr[T = t, I = i|S_{i_k} = -1, S_{i_{<k}} = s_{i_{<k}}]\frac{1}{2}}{Pr[T = t, I = i|S_i = 1, S_{i_{<k}} = s_{i_{<k}}]} \text{ (the bits in } S \text{ are sampled independently uniformly)}$$

$$= \frac{Pr[T = t, I = i|S_{i_k} = -1, S_{i_{<k}} = s_{i_{<k}}]\frac{1}{2}}{\frac{1}{2}Pr[T = t, I = i|S_{i_k} = -1, S_{i_{<k}} = s_{i_{<k}}] + \frac{1}{2}Pr[T = t, I = i|S_{i_k} = 1, S_{i_{<k}} = s_{i_{<k}}]} \text{ (Law of total probability)}$$

$$= \frac{Pr[T = t, I = i|S_{i_k} = -1, S_{i_{<k}} = s_{i_{<k}}]}{Pr[T = t, I = i|S_{i_k} = -1, S_{i_{<k}} = s_{i_{<k}}] + Pr[T = t, I = i|S_{i_k} = 1, S_{i_{<k}} = s_{i_{<k}}]} \text{ (Law of total probability)}$$

$$= \frac{Pr[T = t, I = i|S_{i_k} = -1, S_{i_{<k}} = s_{i_{<k}}]/Pr[T = t, I = i|S_{i_k} = 1, S_{i_{<k}} = s_{i_{<k}}]}{Pr[T = t, I = i|S_{i_k} = -1, S_{i_{<k}} = s_{i_{<k}}]/Pr[T = t, I = i|S_{i_k} = 1, S_{i_{<k}} = s_{i_{<k}}] + 1} \text{ (algebra)}$$

$$\in \left[\frac{e^{-\varepsilon(M)}}{e^{-\varepsilon(M)} + 1}, \frac{e^{\varepsilon(M)}}{e^{\varepsilon(M)} + 1}\right] \text{ (} T \text{ and } I \text{ are functions of } S_{i_{<k}} \text{ and } O\text{)}$$

$$= [1 - p(\varepsilon(M)), p(\varepsilon(M))] \text{ (algebra)}.$$

Hence, the probability of this bit to be 1 conditioned on the same events is bounded in the same way, $Pr[S_{i_k} = 1|T = t, I = i, S_{i_{<k}} = s_{i_{<k}}] \in [1 - p(\varepsilon(M)), p(\varepsilon(M))]$. Therefore, for any guess for the $i$th sampled bit the success probability conditioned on the guesser's outputs and the previously sampled bits is bounded, $Pr[S_i = t_i|T = t, I = i, S_{i_{<k}} = s_{i_{<k}}] \leq p(\varepsilon(M))$. Therefore, if $T_i \neq 0$, $Pr[S_i = t_i|T = t, I = i, S_{i_{<k}} = s_{i_{<k}}] \leq p(\varepsilon(M))$, and since if $T_i = 0$, $S_i \neq T_i$, the random variable $\mathbb{1}_{T_i=S_i}|T = t, I = i, S_{i_{<k}} = s_{i_{<k}}$ is stochastically dominated by $\mathbb{1}_{T_i\neq 0}\text{Ber}(p(\varepsilon(M)))$.

We prove the induction step,

$$V_{k+1}|(T=t) = V_k|(T=t) + \mathbb{1}_{T_i=S_i}|(T=t)$$
$$\preccurlyeq \text{Bin}\left(\|t_{i_{\leq k}}\|_1, p\left(\varepsilon(M)\right)\right) + \mathbb{1}_{T_i \neq 0}\text{Ber}\left(p\left(\varepsilon(M)\right)\right) \text{ (by Lemma E.4)}$$
$$\stackrel{d}{=} \text{Bin}\left(\|t_{i_{\leq k+1}}\|_1, p\left(\varepsilon(M)\right)\right).$$

Therefore, $V|(T=t) \preccurlyeq \text{Bin}\left(\|t\|_1, p\left(\varepsilon(M)\right)\right)$ and hence using A.7, Adaptive ORA is a valid auditing method. $\qquad\square$

## E.3 Adaptive ORA Improves ORA

In Adaptive ORA, the guesser has more information about the other elements, and hence the efficacy increases.

### E.3.1 Xor-in-Pairs

We show the "Xor-in-Pairs" algorithm as an example of an algorithm for which this efficacy gain is maximal. Notice that this efficacy gain varies significantly across different algorithms.

The efficacy of ORA for Xor-in-Pairs is minimal, but Adaptive ORA has perfect efficacy and unlimited guesses with respect to it.

**Definition E.6** (Xor-in-Pairs). For every even $n$, the "Xor-in-Pairs" algorithm $XIP^n : \{0,1\}^n \to \{0,1\}^{\lceil \frac{n}{2} \rceil}$ is the algorithm that groups the elements to pairs by their order and outputs the XOR value of each pair.

$$XIP^n(D) = D_1 \oplus D_2, ..., D_{n-1} \oplus D_n.$$

Similarly to the XOR algorithm, the Xor-in-Pairs is deterministic and it is not differentially private, but ORA is not asymptotically tight for it. In ORA, for every guesser $G$ and element $i \in [n]$, the success probability of $G$ in the $i$th element is $\frac{1}{2}$. However, in Adaptive ORA the optimal accuracy for this algorithm is 1, and hence Adaptive ORA is asymptotically tight for it.

**Proposition E.7.** *For every even $n$, Adaptive ORA is asymptotically tight for $XIP^n$.*

*Proof.* The guesser that guesses the elements by their order, and guesses

$$T_i = \begin{cases} 0 & \text{if } i \text{ is odd} \\ S_{i-1} \oplus O_{\lfloor \frac{i}{2} \rfloor} & \text{otherwise,} \end{cases}$$

that is, abstains from guessing the elements in the odd indexes, and guesses the elements in the even indexes based on the true values of those in the odd indexes, has perfect accuracy and unlimited guesses. $\qquad\square$

### E.3.2 Count-In-Sets

In this section, we discuss auditing of the Count-In-Sets algorithm both with ORA and with AORA. We do not provide a comprehensive analysis of optimal guessers, but rather show an example to demonstrate the advantage of AORA. For this demonstration, we consider auditing with maximum-likelihood guessers that guess only when they have full certainty.

First, consider auditing of the Count algorithm (see Section D.2.1) with guessers that guess only elements for which they have full certainty; that is, a maximum-likelihood guesser with threshold for the absolute value of the privacy loss $\tau = \infty$. All of the taken guesses of these guessers are accurate, and we are interested in the expected number of guesses they take, $\mathbb{E}[R_n]$.

We begin with the non-adaptive ORA. The Count algorithm is symmetric so the distributional privacy loss is the same for all elements, and it is infinite if and only all of the elements $D_1, \ldots, D_n$ have the same value, either 0 or 1, and it happens if and only if $o \in \{0, n\}$ (it can be verified by the proof of

Proposition D.2). Therefore,

$$\mathbb{E}[R_n] = \Pr_{o \sim \mathrm{Bin}\left(n, \frac{1}{2}\right)} [o \in \{0, n\}]$$
$$= 2^{-(n-1)}.$$

Notice that for every $R_1 = 1$, and that it monotonically decreases to $0$.

Now consider Adaptive ORA of Count with a maximum-likelihood guesser that guesses the elements from the last to the first (the order does not matter for symmetric algorithms) with the same threshold $\tau = \infty$, but now over the distributional privacy loss with the distribution conditioned on its observations so far.

When the adaptive guesser guesses the $i$th element it already knows the true values of the previously guessed elements $D_{i+1}, \ldots, D_n$, so it can compute the sum of the remaining elements, and use only this sum for the current guess. Therefore, it accurately guesses the element in the $i$th index if and only if all the elements $D_1, \ldots, D_i$ have the same value, either $0$ or $1$. Therefore, the number of guesses $R_n$ is the largest index $i$ such that $D_1 = \ldots = D_i$. Therefore, $R_n \sim min(\text{Geometric}\left(\frac{1}{2}\right), n)$, and

$$\mathbb{E}[R_n] = \mathbb{E}\left[min\left(\text{Geometric}\left(\frac{1}{2}\right), n\right)\right]$$
$$= \sum_{i=1}^{n} \left(\frac{1}{2}\right)^i \cdot i + \sum_{i=n+1}^{\infty} \left(\frac{1}{2}\right)^i \cdot n$$
$$= 2 - 2^{-(n-1)}.$$

Notice that $R_1 = 1$, and that it monotonically increases to $2$.

Consider auditing of the Count-In-Sets algorithm (see Definition 6.1) with similar maximum-likelihood guessers with $\tau = \infty$. Since the different sets do not affect each other, this algorithm is composed of multiple instances of Count, each with $s(n)$ elements. Therefore, if $s(n) = 1$, both the ORA guesser and AORA guesser have one guess per set in expectation. However, for larger values of $s(n)$, the AORA guesser has more guesses than the ORA guesser. For example, with $s(n) = 10$, the AORA guesser has about $1.998$ expected guesses per set, whereas the ORA guesser has only $0.002$ expected guesses per set. This example highlights the improvement of AORA over ORA for more realistic algorithms.

We ran an empirical simulation of ORA and AORA of Count-In-Sets. Figure 9 displays the results as a function of the size of the sets $s(n)$. We plot the bounds on the privacy level that the auditing method outputs, i.e., $\varepsilon'_\beta(r, v)$ with $\beta = 0.05$ corresponding to a confidence level of $95\%$, with a curve for each auditing method. We use maximum-likelihood guessers with $\tau = \infty$, as described above. The number of sets is fixed to be $100$, and each point represents $100$ experiments.

AORA produces tighter bounds than ORA when $s(n) > 1$. As predicted by the theoretical discussion, both methods have the same results for $s(n) = 1$, but as $s(n)$ increases the AORA bounds tighten while those of ORA loosen. Since setting $\tau = \infty$ yields perfect guessing accuracy, the bounds are fully determined by the statistical power of the number of taken guesses. Therefore, the trend of the bounds exactly matches the trend of the number of taken guesses.

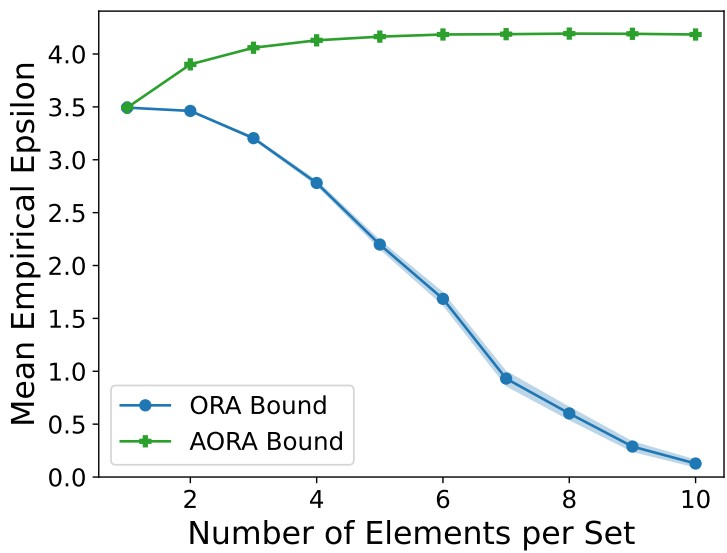

Figure 9: Comparison of the effect of the size of sets $s(n)$ on the results of ORA and AORA of Count-In-Sets. AORA outperforms ORA thanks to its resilience to increased interference.

