# OpenReview forum: "How Well Can Differential Privacy Be Audited in One Run?"
_NeurIPS.cc/2025/Conference — NeurIPS 2025 spotlight_

### Official Review · Reviewer_q21v · 2025-07-02

**Clarity:** 4
**Significance:** 2
**Originality:** 4
**Rating:** 5
**Confidence:** 4

**Summary:**

Differential privacy (DP) provides theoretical upper bounds on information leakage from machine learning models, but real-world implementations can suffer from bugs or overly conservative bounds. This motivates the need for auditing methods to empirically estimate lower bounds on privacy loss.

A promising approach is one-run auditing (ORA)—auditing by running the model once over a dataset with multiple modified entries. This paper addresses a key question left open by Steinke et al. (2024):

How tight can the bounds from ORA be?

Key Contributions:

1. Three Fundamental Gaps that limit ORA’s efficacy:

- Non-worst-case element privacy: Poor privacy for a few elements doesn’t boost overall auditing efficacy.

- Non-worst-case outputs: The algorithm may not produce privacy-violating outputs often enough.

- Interference: Simultaneous changes in multiple data points dilute distinguishable effects.

2. Theoretical Characterization:

- The efficacy of ORA is formally defined and linked to relaxed versions of DP (like average-case and distributional variants).

- Theorem 5.1/5.2: Show how efficacy is bounded below the true ε depending on interference and abstention.

- Theorem 5.3: Gives necessary and sufficient conditions for ORA to be asymptotically tight.

3. New Auditing Strategies:

- Proposes improvements to reduce interference (e.g., allowing multiple elements per model dimension).

- Introduces Adaptive ORA, where guesses are made sequentially and use feedback from prior guesses.

4. Empirical Analysis on DP-SGD:

- Shows empirically that ORA performance initially improves with more auditing elements per dimension, but then degrades due to interference.

In essence, this is a strong theoretical paper with clear motivation, rigorous proofs, and actionable insights for the privacy auditing community.

**Questions:**

- Your theoretical results assume uniform sampling over pairs in the one-run auditing setup. How sensitive are your efficacy/tightness results to this assumption? Could structured or real-world data distributions affect your conclusions?

- You mention a trade-off between statistical power and confidence when allowing abstentions. Have you considered formalizing a method to optimally choose the abstention threshold based on desired confidence bounds or sample budget?

- You cite f-DP auditing work (Mahloujifar et al.), but your efficacy bounds use max-divergence and average-case relaxations. Could similar bounds be derived under f-DP or Rényi DP assumptions?

- Could your results inform DP system designers or regulators in how to audit deployed models? Should one-run auditing be recommended only with specific caveats?

- Have you considered how post-hoc privacy auditing methods under distribution shift might complement one-run auditing, especially for mitigating the non-worst-case output gap?

**Ethical Concerns:**

["NO or VERY MINOR ethics concerns only"]

**Final Justification:**

The authors have sufficiently answered my concerns and also agreed to incorporate my suggestions. I increase my score, this is a technically solid work

**Limitations:**

- Assumes Pure DP: Focuses on ε-DP, not (ε, δ)-DP (though the authors justify it and acknowledge this in Section 1.1 as future work).

- Abstention Strategy Not Fully Executed:
While the theory discusses tradeoffs with abstentions, the practical implications for choosing abstention thresholds in realistic budgets are not deeply explored.

- More in-depth related work would be good. Misses relevant recent work like PANORAMIA, which could have contextualized their limitations better.

**Paper Formatting Concerns:**

Please make sure all cross-references (e.g., “Theorem C.10”) are clearly labeled and described in the main text so reviewers aren't forced to search the appendix with difficulty.

The notation in Section 5 is heavy (e.g., multiple relaxations: AE-AC-DDP, AC-DDP, εkAE-AC), and not always introduced with sufficient intuitive explanation up front. Consider a table summarizing the definitions or including a “notation” box or figure.

The paper refers to Figures 1 and 2, but these appear without proper captions or placement. The figures seem embedded in the LaTeX in a way that breaks up flow and lacks informative captions or labels.

**Quality:**

3

**Strengths And Weaknesses:**

**Strengths:**

- Comprehensive analysis of theoretical and practical limitations of ORA.

- Novel theoretical framework to explain when and why ORA fails.

- Provides actionable auditing improvements, like Adaptive ORA and refined guessing strategies.

- Empirical results align well with theory, adding credibility.

**What’s Related (but not redundant):**
Steinke et al. [1]: Proposed and empirically validated ORA; showed its validity. Mahloujifar et al. [11]: Used f-DP to improve auditing bounds. Xiang et al. [12]: Information-theoretic study of ORA with a focus on interference, but lacks the full characterization and mitigation ideas this paper offers.

What's good is that this is not a duplication of prior work. It is a clear step forward in understanding the limitations and achievable precision of one-run privacy auditing.

Mentioning prior post-hoc privacy auditing frameworks such as PANORAMIA (https://arxiv.org/abs/2402.09477) would strengthen the discussion more, as while it's not a true privacy audit, it offers alternative or complementary strategies to overcome the limitations of one-run auditing under distribution shift or unknown membership settings. I believe it offers a complementary lens to the one-shot vs. multi-shot auditing tradeoff and is worth mentioning.



**Weaknesses and Limitations:**

- Narrow empirical scope: Evaluations are limited to synthetic settings and DP-SGD.

- Exploring real-world datasets or broader ML tasks (e.g., vision/NLP) may improve the quality of the paper for the future, but is not necessary at this point. Again, as future work, the practical implementation of adaptive ORA may help; while proposed in theory and appendices, it has not been demonstrated empirically.

- No exploration of abstention strategies under real-world constraints:

- How would one select a threshold for abstention under budget or privacy constraints?

- Assumption-heavy analysis: Focus on pure ε-DP, assumes knowledge of S vector in ORA, and abstract gradients.

---

> ### Author Rebuttal · Authors · 2025-07-31
>
> Thanks for reading our work and sharing your detailed feedback!
>
> We completely agree that our paper, which maps one-run auditing’s fundamental capabilities and limitations, highlights the importance of future work that will further explore the empirical performance of one-run auditing in a range of real-world settings. While this is out of scope for our paper, there is clearly a lot that could be done.
>
> We would be glad to use some of the one additional page in the camera-ready version to move up some of the results we have on Adaptive One-Run Auditing from the appendix to the main body—on its validity, its maximal efficacy improvement, and on Adaptive ORA of the Count-In-Sets algorithm (the simplified algorithm that captures the essence of auditing DP-SGD). This will help the reader appreciate how AORA is not only immune to the increased interference when adding more elements per coordinate, but actually can only benefit from such added elements. We also have some additional experimental results on Adaptive ORA of DP-SGD (in the same setting as the other DP-SGD experiments) that we omitted from the submission; this felt like too much to try to squeeze into the paper to us, but if you think it is important, we can try to also fit it in.
>
> Thanks for the PANORAMIA reference; we’ll check it out and incorporate it.
>
> We actually don’t think our modeling assumptions (focus on pure DP, case-study with arbitrary gradients) represent a weakness. Since our goal is to expose the limitations of one-run auditing, we intentionally study a setting that is favorable to one-run auditing.
>
> The $S$ vector is randomly sampled to determine the input dataset and the guesser has no access to it, as in the ORA setting that Steinke et al. define.
>
> Thanks for the formatting and notation suggestions—those are well-taken and we’ll incorporate them.
>
> Regarding your specific questions:
>
> - **Data distribution** - We follow the setting of Steinke et al. (“Privacy Auditing with One (1) Training Run”) in which the canaries are sampled independently uniformly. A recent work by Swanberg et al. (“Beyond the Worst Case: Extending Differential Privacy Guarantees to Realistic Adversaries“, Sections 3.1 and 3.2) extends their method to arbitrary product distributions. The gaps that we discuss also exist with this sampling scheme - the guesser still faces non-worst case output, non-worst-privacy for elements, and it still suffers from the uncertainty about the other elements. Therefore, our analysis can be extended to this case, taking into account the non-uniform input distribution.
>
>     Auditing with real data (as canaries or as non-canaries) does not change the sampling (the distribution of the sampled vector S) so our analysis covers this case.
>
> - **Threshold selection** - Choosing the threshold is an interesting problem, and it is a non-trivial task even in classic auditing. However, it is somewhat orthogonal to our focus on the inherent limitations of ORA. As Steinke et al. (“Privacy Auditing with One (1) Training Run”, Section 6) point out, one can use sample splitting to choose an optimal guessing threshold, that is, use a different set of observations to select the threshold and another set of the data for the auditing itself. Formalizing a way to choose the threshold without sample splitting is an interesting research direction.
>
> - **f-DP and Renyi-DP** - We analyze the ORA method of Steinke et al. (“Privacy Auditing with One (1) Training Run”), which audits pure DP and approximate DP. As we discuss in the introduction, we focus on auditing pure DP to zoom in on the limitations of ORA, as auditing approximate DP may create additional gaps. The work of Mahloujifar et al. on auditing f-DP in one run focuses on bridging these additional gaps resulting from the lack of tightness in the approximate DP analysis, and hence doesn’t speak to the pure-DP gaps that our paper studies. Our work shows that average-case relaxations characterize the efficacy of ORA, even though ORA aims to measure the pure DP level, which is characterized by the max divergence.
>
> - **Informing auditing in practice** - Great point, thanks. We will add something on this to the conclusion. Our results indeed can be used to guide a privacy auditor on when to use ORA, how to use it, and how to interpret the results:
>     - ORA should be used only if ORA has high optimal efficacy with respect to the audited algorithm. When auditing algorithms that severely suffer from the gaps of ORA (e.g., an algorithm for releasing noisy counting queries), multi-run auditing should be used.
>     - Our results can guide the choice of the canaries and the guesser, and our multiple canaries per dimension and Adaptive ORA approaches can increase the efficacy of auditing.
>     - Our results highlight that auditing results should not be treated as privacy level estimations, but rather only as lower bounds, and should be interpreted in light of the optimal efficacy.
>
>     When using ORA, we recommend reporting the number of canaries used, the optimal efficacy of ORA with respect to the audited algorithm, and other details which are important for the interpretation of the results (aligned with the “Auditing Card” approach proposed by Annamalai et al. (“The Hitchhiker’s Guide to Efficient, End-to-End, and Tight DP Auditing”, Section 7.3).
>
> - **Auditing under distribution shift** - We have not considered post-hoc privacy auditing methods under distribution shift. We will explore this complementary approach.

---

> ### Comment · Reviewer_q21v · 2025-08-04
>
> Thanks so much for answering my questions and incorporating the suggestions esp regarding related work and distribution shifts use case. I have increased my score.

---

### Official Review · Reviewer_UhJV · 2025-07-02

**Clarity:** 3
**Significance:** 4
**Originality:** 4
**Rating:** 5
**Confidence:** 3

**Summary:**

The paper addresses the problem related to how accurately one can audit differential-privacy guarantees from a single execution of an algorithm, a technique called One-Run Auditing (ORA). The authors show that ORA’s precision is fundamentally limited by three sources of errors: 1) non-worst-case elements; 2):non-worst-case outputs; 3) interference among the many perturbed records injected at once. In this paper, the authors attempt to (i) derive tight upper bounds on the best accuracy any ORA scheme can reach, (ii) present necessary and sufficient conditions for ORA to become approximately tight, and (iii) demonstrate how these limits are achieved for Differentially Private Stochastic Gradient Descent (DP-SGD). They further propose practical tweaks, such as planting several canary elements per gradient dimension and an Adaptive ORA that refines its guesses iteratively, regaining some of the accuracy lost to interference. Overall, the work rigorously advances our understanding of the inherent limits and attainable power of privacy auditing.

**Questions:**

1. How would ORA behave on realistic DP-SGD workloads—e.g. CIFAR-10, Fashion-MNIST, AG-News, or Purchase-100—evaluated at privacy budgets typical of deployments? Do you plan to add such experiments, or can you argue why the current synthetic setting is sufficient to establish generality?
2. Can you quantify, on full-scale models, the accuracy gains and computational costs of your two heuristics (multiple canaries per gradient dimension and Adaptive ORA)? A small-scale demonstration would help readers gauge their real-world feasibility.

**Ethical Concerns:**

["NO or VERY MINOR ethics concerns only"]

**Final Justification:**

The authors have addressed my concerns, and I am comfortable changing my rating.

**Limitations:**

The authors run gradient-canary audits on a 1,000-dimensional DP-SGD synthetic dataset. Experiments on larger datasets would have been more interesting.

**Paper Formatting Concerns:**

Not much of concern.

**Quality:**

3

**Strengths And Weaknesses:**

Strenghts

1. The paper clearly explains its main idea through the gap decomposition and inequalities (From Theorems 5.1–5.2), offering a simple way to understand ORA. Theorem 5.3 clearly identifies when ORA matches traditional auditing, resolving a previously open issue; for example, the question of whether a single-run audit could ever achieve the same asymptotic accuracy as traditional multi-run differential-privacy auditing, given that empirical results often show that ORA often fails to do so in practice.

2. The authors present a nice case study on auditing DP-SGD with one-run auditing (ORA) and pinpoint how its gradient aggregation produces a severe interference gap that muffles privacy signals. Then the authors embed multiple canary gradients per coordinate to increase high-signal elements, and deploy an Adaptive ORA that conditions guesses made at later stages on earlier ones to further cut through interference. This approach is interesting.

3. Overall, the paper delivers a complete theoretical account of one-run auditing: it proves that three precisely-defined gaps bound the best efficacy any ORA method can ever reach (Theorems 5.1–5.2) and pinpoints the exact condition: an unbounded supply of near–worst-case elements, under which ORA attains the same asymptotic tightness as multi-run audits (Theorem 5.3). It then shows how these limits manifest for DP-SGD and proposes two concrete mitigations (multiple canaries per dimension and Adaptive ORA) to narrow the interference gap.

Weaknesses

1. There are some limitations on experiments; tests are done on a simple 1,000-dimensional synthetic model. If the author could include more realistic scenarios (vision, NLP, lower-dimensional tabular models) with practical privacy levels, it would further strengthen the results. For stronger external validity, it would have been nice to see real-world DP-SGD workloads: e.g., Fashion-MNIST or CIFAR-10 (Jagielski et al., 2020; Nasr et al., 2023), Purchase-100 (Jagielski et al., 2020; Nasr et al., 2023), or AG-News text classification (Steinke et al., 2023) evaluated at realistic privacy budgets. Extending such experiments further would enhance the scalability and practicality of the solution.

2. Adaptive ORA is briefly analyzed, with minimal experimental validation, even though it’s highlighted as an important improvement.

3. The authors characterize in Theorem 5.3 that ORA is tight whenever “the number of elements whose |l| is arbitrarily close to \epsilon is unbounded”, and the authors mention the idea on DP-SGD with two heuristics: 1. adding multiple canaries per gradient dimension; 2. the adaptive ORA variant. But they state that their experiments are meant only to “illustrate theoretical insights … not to serve as a comprehensive evaluation in realistic settings” and state that the entire audit “can be executed on a standard laptop (we do not train ML models)”. So while the theoretical aspects are strong, the practical realization is somewhat unclear/uncertain.

4. Possible ways authors can further enhance the paper without losing rigor:
  (1) Perhaps the authors can use an overview diagram that shows how the three gaps map to their respective relaxed privacy notions and inequalities.
 (2): Proof relocation: it is perhaps enough to keep high-level proof sketches in the main text, and move algebraic or complicated details to the appendix. This would enhance the readability of the paper for a broader audience.
 (3): It would be helpful to collect all symbols and their meanings in one place so readers do not search across sections.

5. The analysis is limited to pure \epsilon-DP. Because most deployed systems use (\epsilon, \delta)-DP it would help to either (i) extend Theorems 5.1–5.3 to the approximate setting, or (ii) empirically quantify how the \delta-tail degrades auditing bounds. Given its absence, the practical relevance of the theory remains uncertain.

---

> ### Author Rebuttal · Authors · 2025-07-31
>
> Thank you for taking the time to read our work and share your feedback!
>
> We completely agree that our paper, which maps one-run auditing’s fundamental capabilities and limitations, highlights the importance of future work that will further explore the empirical performance of one-run auditing in a range of real-world settings. While this is out of scope for our paper, we think you lay a good roadmap for what some of that future work could do.
>
> We also agree that there is room for future research extending our formal results to approximate DP. That said, our results should already have strong relevance to the approximate DP setting, since for reasonable values of delta, the reasons for the gaps still apply and similar results will hold. For example, consider the Name and Shame algorithm that randomly selects an element and outputs it. When applying ORA, the auditor can guess only one element better than random, and hence for any reasonable value of $\delta << \frac{1}{n}$ that one sets as input for auditing, ORA would discover a bound on epsilon close to $0$ (despite the true value being infinity). More generally, for very small delta, an $(\epsilon, \delta)$-DP algorithm "most of the time’’ must behave "similarly" to an $(\epsilon, 0)$-DP algorithm with respect to any element of its input, which doesn’t leave much opportunity for ORA to leverage.
>
> On the empirical side, in our DP-SGD experiments we set $\delta=10^{-5}$. We note that for this delta, with the number of elements and guesses that we use, the "fixing term" (derived by Steinke et al. in "Privacy Auditing with One (1) Training Run") that adjusts the epsilon-DP bounds of ORA to account for delta is negligible. For example, with $1,000$ elements, and $70$ accurate guesses out of $100$ taken guesses, the difference between the resulting epsilon bounds (accounting for the delta vs. not accounting for the delta) is approximately $2.5 \cdot 10^{-3}$. In addition, Steinke et al. (Figure 12) have empirically explored the effect of delta on their auditing results. Their results align with our observation that when $\delta << \frac{1}{n}$, the effect of delta is negligible.
>
> Thanks for the concrete suggestion to include more on Adaptive One-Run Auditing in the main body of the paper. We would be glad to use some of the one additional page in the camera-ready version to do exactly this, moving up some of the results we have in the appendix on its validity, its maximal efficacy improvement, and on Adaptive ORA of the Count-In-Sets algorithm (the simplified algorithm that captures the essence of auditing DP-SGD). This will help the reader appreciate how AORA is not only immune to the increased interference when adding more elements per coordinate, but actually can only benefit from such added elements. We also have some additional experimental results on Adaptive ORA of DP-SGD (in the same setting as the other DP-SGD experiments) that we omitted from the submission; this felt like too much to try to squeeze into the paper to us, but if you think it is important, we can try to also fit it in.
>
> We liked your suggestions for further enhancement and are working on implementing them. We actually already created such a diagram in slides on this work, and agree that it would be nice to include it in the paper. We agree that more algebraic details can be pushed to the appendix to improve flow and free up space for other content that is more essential to the main body. And a table of symbols is also a nice idea.

---

> > ### Comment · Reviewer_UhJV · 2025-08-04
> >
> > Thanks a lot for your clarification. I appreciate that you have considered updating your paper with additional A-ORA results and other minor suggested improvements. Still not fully convinced about the generalizability of the "negligible" comment related to delta. However, at least making sure these are adequately discussed would encourage me to reconsider my rating (to a higher one). Thanks.

---

> > > ### Author Response · Authors · 2025-08-06
> > >
> > > Thanks for your response! We agree that a more comprehensive discussion of approximate-DP will strengthen the paper, and we will make sure that the camera-ready version includes one.
> > >
> > > In this discussion, we will explain why we expect our pure-DP results to remain relevant for approximate DP, as long as the values of delta are reasonable. The essence of the argument is that an $(\epsilon, \delta)$-DP algorithm must “most of the time” behave “similarly” to an $(\epsilon, 0)$-DP algorithm, and hence the gaps we identify still apply - and the extension of ORA to approximate DP only creates additional gaps. We will support this argument with the result by Kasiviswanathan and Smith (“On the ‘Semantics’ of Differential Privacy: A Bayesian Formulation,” Lemma 3.3, Part 2) and illustrate it with the Name and Shame example. We will also reference the discussion by Steinke et al. (“Privacy Auditing with One (1) Training Run,” Section 7) and existing empirical results.
> > >
> > > We hope that this will better justify our pure-DP focus while acknowledging the importance of future research on approximate DP auditing. Thanks for the helpful suggestion! We hope that this change will make you comfortable increasing your score.

---

> > > > ### Comment · Reviewer_UhJV · 2025-08-08
> > > >
> > > > Thank you for your further response. With those changes ensured, I am happy to upgrade my rating.

---

### Official Review · Reviewer_YJxk · 2025-07-03

**Clarity:** 3
**Significance:** 3
**Originality:** 3
**Rating:** 5
**Confidence:** 3

**Summary:**

This paper analyzes the capabilities and limitations of one-run auditing (ORA), a method for estimating the privacy loss of differentially private algorithms using a single execution. The authors identify three key obstacles to ORA's accuracy—non-worst-case elements, non-worst-case outputs, and interference between multiple perturbed inputs—and formally characterize their impact on auditing efficacy. They show that ORA is asymptotically tight only under specific conditions, such as for local algorithms. A case study on DP-SGD illustrates these limitations and proposes new strategies, such as multi-element assignments per dimension and adaptive ORA, to further improve auditing performance.

**Questions:**

The paper is well-written, and the technical details are clearly presented. I have no further questions.

**Ethical Concerns:**

["NO or VERY MINOR ethics concerns only"]

**Final Justification:**

I will keep my positive score

**Limitations:**

yes

**Paper Formatting Concerns:**

No formatting concerns as far as I know.

**Quality:**

3

**Strengths And Weaknesses:**

**Strengths**
The paper provides a thorough theoretical analysis of the efficacy of one-run auditing (ORA) and rigorously identifies three key scenarios where ORA fails to be asymptotically tight. These limitations are formalized with clear proofs. Additionally, the paper proposes Adaptive ORA and multi-element assignments to improve ORA's performance, supported by both theoretical justification and empirical results.

**Weaknesses**
The evaluation is limited to the white-box setting. Extending the experiments to black-box scenarios and comparing performance would offer a more comprehensive understanding of ORA’s practical applicability and the advantage of the two proposed methods.

---

> ### Author Rebuttal · Authors · 2025-07-31
>
> Thank you for taking the time to read our work and share your feedback!
>
> We agree that, following on from this paper, larger-scale empirical studies of one-run auditing, including in the black-box setting, are an important avenue for future work.
>
> A few brief thoughts on these directions:
> - We expect that ideas we have explored in the white-box auditing model will also prove relevant in the black-box model. For example, similar phenomena when increasing the number of canaries were empirically observed in the black-box setting by Steinke et al. Furthermore, our theoretical analysis of the sum of the gradients should also be relevant for black-box auditing. (Black-box membership inference attacks using the gradient [1] or Hessian [2] at the final model are known to be stronger than attacks that use only the model predictions [1, 2], and under the KKT conditions the model can be represented as a weighted sum of the gradients of the loss w.r.t. the training points [3].)
> - A thorough investigation of Adaptive ORA for the black-box setting will require the development of more sophisticated black-box attacks designed to exploit partial knowledge about other elements, a challenging (and fascinating) future direction.
>
>
> [1] Pang, Yan, et al. "White-box Membership Inference Attacks against Diffusion Models." Proceedings on Privacy Enhancing Technologies (2025).
>
> [2] Suri, Anshuman, Xiao Zhang, and David Evans. "Do Parameters Reveal More than Loss for Membership Inference?" Transactions on Machine Learning Research.
>
> [3] Smorodinsky, Guy, Gal Vardi, and Itay Safran. "Provable Privacy Attacks on Trained Shallow Neural Networks." arXiv preprint arXiv:2410.07632 (2024).

---

> ### Comment · Reviewer_YJxk · 2025-08-06
>
> Thanks for answering my question. I will keep my positive score.

---

### Official Review · Reviewer_pJ8q · 2025-07-03

**Clarity:** 2
**Significance:** 3
**Originality:** 3
**Rating:** 5
**Confidence:** 3

**Summary:**

The authors investigate the challenge of auditing the privacy guarantees of a mechanism using just one training run. This approach has gained popularity in recent years because traditional auditing methods, which require running the mechanism multiple times, can be computationally expensive. While single-run auditing makes the process more cost-effective, it also comes with the trade-off of potentially weaker lower bounds on the privacy estimates. This means practitioners may not always have a clear understanding of how accurate these estimates are. The paper sheds light on the factors that contribute to these weaker bounds and introduces the concept of "efficacy" to analyze the accuracy of privacy estimates in popular mechanisms like DP-SGD. They identify three main factors that contribute to these weaker lower bounds: interference between canaries, the presence of non-worst-case canaries, and non-worst-case outputs.

**Questions:**

The authors mention that the Dirac canary introduced by Nasr et al. is particularly effective at minimizing interference between canaries. However, I wonder if this advantage is as significant as suggested. In high-dimensional settings, which are typical for DP-SGD, the inner product between a random canary and the noise-added batch gradient is likely to be close to zero with high probability, leading to minimal interference. This would also apply to random canaries, not just Dirac canaries. Could the authors elaborate on the specific advantage of using Dirac canaries over random canaries in this context?

**Ethical Concerns:**

["NO or VERY MINOR ethics concerns only"]

**Final Justification:**

After the rebuttal phase, I remain at my score. I believe that this paper produces meaningful new insights into an important problem (privacy auditing in a single run).

**Limitations:**

yes

**Paper Formatting Concerns:**

no concerns

**Quality:**

3

**Strengths And Weaknesses:**

Strengths:

One of the key strengths of this paper is that it addresses a highly important and timely problem. As single-run auditing methods become increasingly popular among practitioners due to their efficiency, understanding the limitations of these methods is crucial. The paper provides valuable insights into when and why these privacy estimates may be less trustworthy, helping practitioners make more informed decisions. Another strength lies in the paper’s introduction of the concept of "efficacy," which is explored with a solid level of mathematical rigor. Despite the theoretical depth, the authors also ensure the work remains highly applicable by focusing on practical mechanisms like DP-SGD, demonstrating the real-world relevance of their findings.

Weaknesses:

My main concern about the paper lies in its presentation. The authors attempt to condense a substantial amount of information into the limited nine-page format, which unfortunately impacts readability. Essential concepts are often introduced in the appendix, resulting in numerous forward references that disrupt the flow of the main text. For instance, in Section 6.1, the function s(n), which plays a central role in that part of the analysis, is not clearly defined within the main body of the paper. While the constraints of the conference page limit are understandable, the heavy reliance on the appendix for critical definitions and explanations can hinder the reader's ability to fully grasp the material in a seamless manner.

---

> ### Author Rebuttal · Authors · 2025-07-31
>
> Thank you for taking the time to read our work and share your feedback!
>
> Your point about the challenges of squeezing so much content into the conference format is well-taken. Thanks to your comment, we have been working on streamlining the content of the main body so that the reader will not need to turn to the appendix for essential definitions. (For example, we have moved the definition of the “Count-In-Sets” algorithm (including the $s(n)$ notation) to the main text.) These changes have significantly improved readability, so thanks!
>
> You are correct that in the typical high-dimensional settings, the interference between random canaries is small. Dirac canaries simply completely eliminate this interference and therefore have a small advantage (as studied by Nasr et al. in their Appendix B.1), and hence we (and Steinke et al.) focus on this model. It’s a nice point, though, and we have added a brief mention of it. The approaches we introduce and the phenomena we analyze, such as the tradeoff when increasing the number of elements beyond the dimension, generalize beyond DP-SGD and beyond this specific attack, and will have similar consequences if one were to use random canaries in a high-dimensional setting.

---

> > ### Comment · Reviewer_pJ8q · 2025-08-04
> >
> > Thank you for your response. I maintain my initial assessment that this is a technically solid paper.

---

### Decision · Program_Chairs · 2025-09-17

**Decision:**

Accept (spotlight)

**Comment:**

The reviewers all agree that this paper provides new insights into the problem of auditing differential private algorithms. The results focus on a recent paradigm called "one-run privacy auditing," which has become popular in the differential privacy literature. The fundamental barriers highlighted in this paper can point to new directions for privacy auditing.